# K-Gen: Unlocking Large-scale Data-Free Knowledge Distillation via Key Region Generation

## Abstract

Data-Free Knowledge Distillation (DFKD) is an advanced technique that enables knowledge transfer from a teacher model to a student model without relying on original training data. While DFKD methods have achieved success on smaller datasets like CIFAR10 and CIFAR100, they encounter challenges on larger, high-resolution datasets such as ImageNet. A primary issue with previous approaches is their generation of synthetic images at high resolutions (e.g., $224 \times 224$) without leveraging information from real images, often resulting in noisy images that lack essential class-specific features in large datasets. Additionally, the computational cost of generating the extensive data needed for effective knowledge transfer can be prohibitive. In this paper, we introduce Key Region Data-free Generation (K-Gen) to address these limitations. K-Gen generates only key region of images at lower resolutions while using class-activation score to ensure that the generated images retain critical, class-specific features. To further enhance model diversity, we propose multi-resolution generation and embedding diversity techniques that strengthen latent space representations, leading to significant performance improvements. Experimental results demonstrate that K-Gen achieves state-of-the-art performance across both small-, high- and mega-resolution datasets, with notable performance gains of up to two digits in nearly all ImageNet and subset experiments. Code is available at `https://anonymous.4open.science/r/K-Gen-DFKD`.

## 1 Introduction

Knowledge distillation (KD) Hinton et al. (2015); Zhao et al. (2022) is a technique aimed at training a student model to replicate the capabilities of a pre-trained teacher model. Over the past decade, KD has been applied across various domains, including image recognition Qiu et al. (2022), speech recognition Yoon et al. (2021), and natural language processing Sanh et al. (2019). Traditional KD methods typically assume that the student model has access to all or part of the teacher's training data. However, in many real-world scenarios, particularly in privacy-sensitive fields like healthcare, accessing the original training data is not feasible due to legal, ethical, or proprietary constraints. In such cases, conventional KD methods become impractical, necessitating alternative approaches that do not rely on direct access to the original data.

To address this challenge, Data-Free Knowledge Distillation (DFKD) Yin et al. (2020); Fang et al. (2021); Yu et al. (2023); Tran et al. (2024a); Patel et al. (2023); Do et al. (2022); Binici et al. (2022a) has emerged as a promising solution. DFKD transfers knowledge from a teacher neural network ($\mathcal{T}$) to a student neural network ($\mathcal{S}$) by generating synthetic data instead of using the original training data. This synthetic data enables adversarial training between the generator and the student Nayak et al. (2019); Micaelli & Storkey (2019), where the student aims to match the teacher's predictions on the synthetic data, while the generator's objective is to create samples that maximize the discrepancy between the teacher's and student's predictions.

Previous works Yin et al. (2020); Fang et al. (2021); Yu et al. (2023); Tran et al. (2024a); Patel et al. (2023); Tran et al. (2024b) typically generate synthetic data at the same resolution as the images used to train the teacher model, a technique that has proven effective on small-scale and

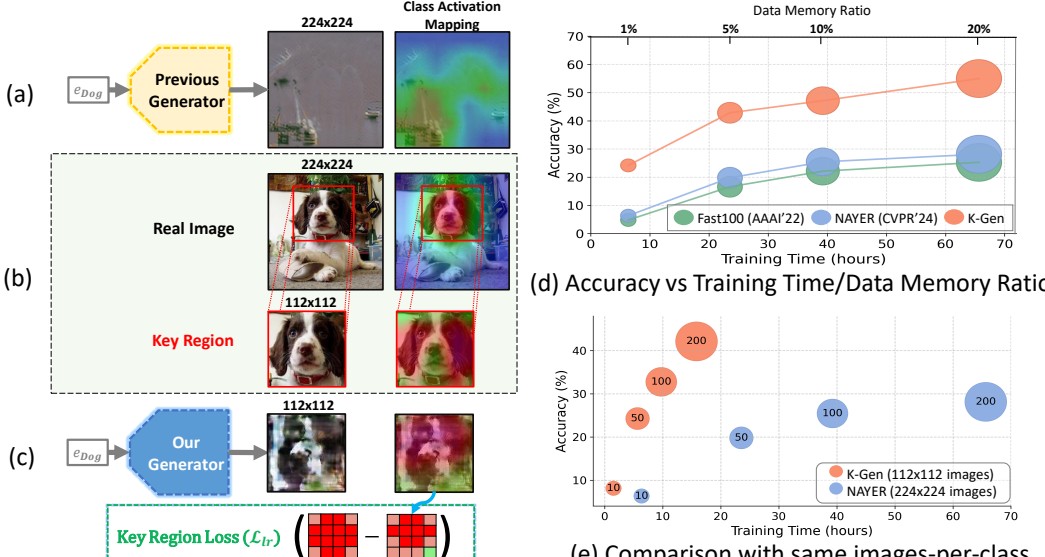

Figure 1: (a) The previous model fails to capture class-specific features and contains a lot of noisy pixels. (b) The visualization demonstrates that only a small set of key features is important for classifiers. (c) Our model generates synthetic images at lower resolutions and leverages CAM to generate pixels in key region, which contains important information. Comparison of K-Gen and SOTA methods on ImageNet1K: (d) performance vs. training time and data memory ratio (note that the training time is positively correlated with the data memory ratio); (e) Performance of K-Gen and NAYER Tran et al. (2024b) with the same image-per-class constraints.

low-resolution datasets like CIFAR10 and CIFAR100 Tran et al. (2024b); Fang et al. (2022). However, these approaches face significant challenges when applied to larger, high-resolution datasets such as ImageNet. A primary issue with previous methods is their generation of synthetic images at high resolutions (e.g., $224 \times 224$) without incorporating information from real images, leading to substantial noise and a lack of nuanced, class-specific features critical for effective knowledge transfer. Additionally, the computational cost of generating the large volumes of synthetic data required for knowledge transfer can be prohibitively high. For instance, previous methods Yin et al. (2020) have demanded over 3,000 GPU hours to train on ImageNet1k, yet have achieved only moderate results. As a result, while DFKD methods perform well on smaller datasets, they encounter substantial limitations when scaled to real-world, large-scale applications.

In this paper, we introduce Key Region Data-free Generation (K-Gen) to tackle the limitations of traditional DFKD methods. Inspired by the observation that only a small but crucial region of real images is essential for effective classifier training Zhu et al. (2020); Selvaraju et al. (2017), K-Gen introduces a strategy that synthesizes lower-resolution images while leveraging Class Activation Scores Zhou et al. (2016) to focus on the most informative pixels. By concentrating on the most relevant areas, K-Gen ensures that the generated images retain critical class-specific features, thereby improving the efficiency of knowledge transfer. Additionally, this approach helps reduce computational costs, enhancing both the scalability and performance of DFKD, especially for large, high-resolution datasets.

As shown in Figure 1a-c, the previous DFKD model generates $224 \times 224$ resolution images, which are often noisy and provide limited information for training the classifier. In contrast, our method produces lower-resolution images that leverage a key region loss to retain discriminative features. Moreover, as illustrated in Figure 1d-e, our method not only significantly speeds up training time but also achieves improved accuracy. Specifically, under the same images-per-class setting and despite generating lower-resolution images, our K-GEN still achieves better performance, demonstrating both its efficiency and effectiveness.

Although using lower-resolution synthetic images improves training efficiency, it may limit the model's capacity to capture diverse and detailed feature representations, as lower resolutions constrain the available representational space. To overcome this limitation, we propose a **Multi-Resolution Data Generation** strategy, in which images are generated at multiple resolutions to capture both coarse and fine-grained features. In addition, we introduce an **Embedding Diversity Loss** to preserve distinctiveness within the latent space, ensuring that rich feature representations are

maintained even at lower resolutions. Together, these mechanisms enable the model to retain critical features across different levels of granularity, leading to enhanced performance and robustness across a variety of tasks.

Our major contributions are summarized as follows:

- We propose Key Region Data-free Generation (K-Gen), which generates synthetic images at lower resolutions, using Class Activation Maps to focus on critical regions, improving computational efficiency without sacrificing essential class-specific features.
- We also extend this to use for Vision Transformer architecture.
- We introduce Multi-Resolution Data Generation to capture both coarse and fine features and Embedding Diversity Loss to maintain distinct embeddings at lower resolutions, boosting feature diversity and performance.
- K-Gen achieves state-of-the-art performance on both low- and high-resolution datasets, including CIFAR10, CIFAR100, TinyImageNet, ImageNet, ImageNet subsets. Our method demonstrates performance gains of up to two digits in nearly all experiments on ImageNet and its subsets.
- K-Gen exhibits high performance on mega-resolution datasets (images >1M pixels), significantly expanding the applicability of DFKD methods to ultra-high-resolution domains.

## 2 RELATED WORK

**Data-Free Knowledge Distillation.** DFKD methods Yin et al. (2020); Fang et al. (2021); Yu et al. (2023); Do et al. (2022); Patel et al. (2023) generate synthetic images to facilitate knowledge transfer from a pre-trained teacher model to a student model. These synthetic data are used to jointly train the generator and the student in an adversarial manner Micaelli & Storkey (2019). Specifically, the student aims to make predictions that closely align with the teacher's on the synthetic data, while the generator strives to create samples that match the teacher's confidence while also maximizing the mismatch between the student's and teacher's predictions. This adversarial process fosters a rapid exploration of synthetic distributions that are valuable for knowledge transfer between the teacher and the student.

**Data-Free Knowledge Distillation for High-Resolution Dataset.** Data-free knowledge distillation methods face significant challenges when scaled to larger, high-resolution datasets like ImageNet. For instance, DeepInv Yin et al. (2020) required over 3000 NVIDIA V100 GPU hours to train on ImageNet1k, highlighting the substantial computational demands. Although more recent methods Tran et al. (2024b); Fang et al. (2022) provide faster solutions, they cannot achieve competitive performance when training models from scratch without the pretrained data used by DeepInv. Therefore, there is an urgent need for novel methods that can efficiently and effectively enable data-free transfer on high-resolution datasets like ImageNet.

## 3 PROPOSED METHOD

### 3.1 PRELIMINARIES: DFKD FRAMEWORK

Consider a training dataset $D = \{(\boldsymbol{x}_i, \boldsymbol{y}_i)\}_{i=1}^{m}$, where each $\boldsymbol{x}_i \in \mathbb{R}^{c \times h \times w}$ is an input sample and $\boldsymbol{y}_i \in \{1, 2, \ldots, K\}$ denotes its label. Each pair $(\boldsymbol{x}_i, \boldsymbol{y}_i)$ in $D$ serves as a training example with its corresponding label. Let $\mathcal{T}$ with parameters $\theta_{\mathcal{T}}$ represent a pre-trained teacher network on $D$. The objective of DFKD is to train a student network, $\mathcal{S} = \mathcal{S}_{\theta_{\mathcal{S}}}$, to match the teacher's performance without access to the original dataset $D$.

To achieve this, inspired by Tran et al. (2024b), we begin by sampling a batch of random pseudo-labels $\hat{\boldsymbol{y}} \sim \{1, \ldots, K\}$. We then obtain their corresponding text embeddings using a pre-trained language model $\mathcal{C}$, i.e., $\boldsymbol{f_y} = \mathcal{C}(\hat{\boldsymbol{y}})$. These embeddings $\boldsymbol{f_y}$ are passed through a noisy layer $\mathcal{Z}$, which is a single linear layer re-initialized at each iteration to introduce randomness and promote diverse image generation. The output is then fed into a lightweight generator $\mathcal{G}$ to produce synthetic images $\hat{\boldsymbol{x}}$.

$$\hat{\boldsymbol{x}} = \mathcal{G}_{l \times l}(\mathcal{Z}(\boldsymbol{f_y})), \tag{1}$$

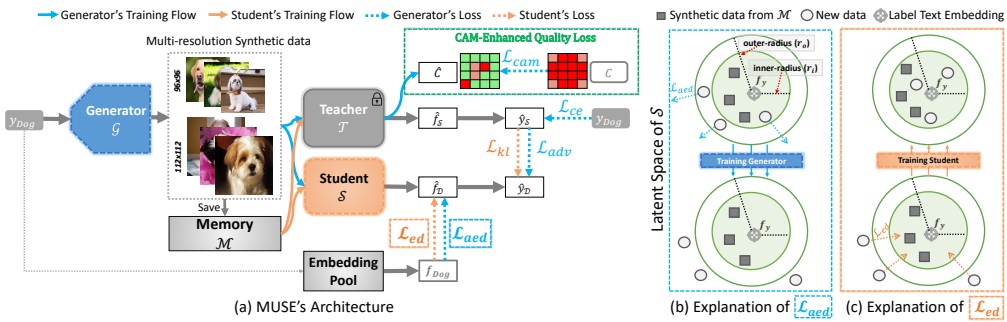

Figure 2: (a) Overview of the K-Gen architecture, illustrating the two-phase training process: generator training and student training. The model generates lower-resolution images and enhances their quality using Key Region Loss, while also promoting diversity through Embedding Diversity Loss ($\mathcal{L}_{ed}$ and $\mathcal{L}_{aed}$). (b) $\mathcal{L}_{ed}$ (Eq. 9) aims to learn the embedding in $\mathcal{S}$ of all old data, bringing it closer to $f_y$, while (c) $\mathcal{L}_{aed}$ (Eq. 11) guides the generator $\mathcal{G}$ to produce new data that is distant from $f_y$, thus enhancing the model's diversity.

where $\hat{x} \in \mathbb{R}^{3 \times l \times l}$, with $l$ representing the resolution of the training data (e.g., $224 \times 224$ for ImageNet or $32 \times 32$ for CIFAR10/CIFAR100). Note that we use $\mathcal{G}_{l \times l}$ to specify the generator that produces $l \times l$ resolution images. Subsequently, $\hat{x}$ is stored in a memory pool $\mathcal{M}$ and used to jointly train both the generator and the student network in an adversarial setup Micaelli & Storkey (2019). In this setup, the student is trained to approximate the teacher's predictions on synthetic data by minimizing the Kullback-Leibler (KL) divergence loss between $\mathcal{T}(\hat{x})$ and $\mathcal{S}(\hat{x})$.

$$\hat{y}_\mathcal{S} = \mathcal{S}(\hat{x}); \quad \hat{y}_\mathcal{T} = \mathcal{T}(\hat{x}) ,$$
$$\mathcal{L}_\mathcal{S} = \mathcal{L}_{\text{KL}} = \text{KL}(\hat{y}_\mathcal{T}, \hat{y}_\mathcal{S}), \tag{2}$$

while the generator aims to produce samples that not only align with the teacher's confidence but also maximize the discrepancy between the student's and teacher's predictions.

$$\mathcal{L}_\mathcal{G} = \alpha_{ce}\mathcal{L}_{\text{CE}}(\hat{y}_\mathcal{T}, \hat{y}) - \alpha_{adv}\text{KL}(\hat{y}_\mathcal{T}, \hat{y}_\mathcal{S}) + \alpha_{bn}\mathcal{L}_{\text{BN}}(\mathcal{T}(\hat{x})). \tag{3}$$

In this framework, $\mathcal{L}_{\text{CE}}$ represents the Cross-Entropy loss, training the student on images within the teacher's high-confidence regions. In contrast, the negative $\mathcal{L}_{adv}$ term encourages exploration of synthetic distributions, enhancing knowledge transfer from the teacher to the student. Here, the student network acts like a discriminator in GANs, guiding the generator to produce images that the teacher has mastered but the student has yet to learn, thereby focusing the student's development on areas where it lags behind the teacher. Additionally, we apply batch norm regularization ($\mathcal{L}_{\text{BN}}$) Yin et al. (2020); Fang et al. (2022), a standard DFKD loss, to align the mean and variance at the `BatchNorm` layer with its running mean and variance. This adversarial setup facilitates the efficient exploration of synthetic distributions for effective knowledge transfer between the teacher and the student.

In comparison with previous works, our method first proposes generating key region data generation at a lower resolution, which synthesize data with high class activation score (Section 3.2). Next, we introduce two techniques to further improve the diversity of our models (Section 3.4). Finally, the overall process is summarized in Section 3.5.

## 3.2 KEY REGION DATA-FREE GENERATION AT LOWER-RESOLUTION

A major limitation of previous approaches is their generation of synthetic images at high resolutions ($224 \times 224$) without incorporating information from real images. This leads to images with significant noise, lacking the class-specific features essential for effective knowledge transfer, as illustrated in Figure 1a-c.

**Key Region Lower-Resolution Data Generation.** To address these limitations, we propose generating synthetic images at lower resolutions.

$$\hat{x} = \mathcal{G}_{3 \times e \times e}(\mathcal{Z}(f_y)), \tag{4}$$

where $\hat{x} \in \mathbb{R}^{3 \times e \times e}$ and $e$ is the target resolution (i.e., $e \ll l$).

To ensure that the synthetic images $\hat{x}$ capture important information, we propose maximizing their CAM with the target map, which contains high values of class activation. First, we use the classic CAM method Zhou et al. (2016) to generate the matrix $M(\hat{x}, \hat{y})$ for the image $\hat{x}$ and class $\hat{y}$:

$$M(\hat{x}, \hat{y}) = \sum_k w_k^{\hat{y}} \mathcal{T}_k(\hat{x}, \hat{y}), \tag{5}$$

where $w_k^{\hat{y}}$ is the $k^{\text{th}}$ weight in the final classification head for class $\hat{y}$, and $\mathcal{T}_k$ is the $k^{\text{th}}$ feature map in the final layers of the model. Note that we only use the latent matrix of CAM, which is before the normalization and interpolation into full-resolution images. Then, the loss function $\mathcal{L}_{\mathcal{G}}$ is modified with additional key region loss ($\mathcal{L}_{kr}$) as follows:

$$\mathcal{L}_{\mathcal{G}} = \alpha_{ce}\mathcal{L}_{ce} + \alpha_{adv}\mathcal{L}_{adv} + \alpha_{bn}\mathcal{L}_{bn} + \alpha_{kr}\mathcal{L}_{kr},$$
$$\mathcal{L}_{kr} = \sum_{h,w \in M} (\max\{0, M_{\text{target}} - M(\hat{x}, \hat{y})\}). \tag{6}$$

In this context, $M_{\text{target}}$ is a predefined mask with high values at the center and lower values at the borders, it like a Gaussian centered on the image, guiding the generator to produce the desired activation map $M(\hat{x}, \hat{y})$. We conducted an ablation study in the appendix G demonstrating that the Gaussian mask with maximum value at 1 and a standard deviation of 2 yield the best results.

By using a margin loss to define $\mathcal{L}_{kr}$, we encourage the values in $M(\hat{x}, \hat{y})$ to *only sufficiently exceed* those in $M_{\text{target}}$, avoiding excessively high values that could negatively impact image quality while concentrating the important values of $M(\hat{x}, \hat{y})$ near the center. Finally, the sum of all pixel values in the tensor is used as $\mathcal{L}_{kr}$.

Thanks to the use of lower-resolution images with key region loss, as shown in Figure 1c, generating lower-resolution images improves accuracy by enabling the generator to capture critical features more effectively. Figure 1d further illustrates the substantial reduction in training time, highlighting the efficiency gains of this approach. Together, these findings underscore the advantages of low-resolution synthetic images in enhancing both performance and computational efficiency in DFKD for high-resolution datasets. For example, with only 9 hours of training, our K-Gen achieves 24.25% accuracy, significantly outperforming DeepInv, which reaches only 3.15% after 61.2 hours of training.

### 3.3 KEY REGION GENERATION FOR VISION TRANSFORMER

A key challenge in our approach is training the student model with lower-resolution images, which are then tested on full-resolution data. This is particularly challenging for patch-based models, such as Vision Transformer (ViT) and its variants Dosovitskiy (2020); Touvron et al. (2021), that do not rely on CNN architectures. Furthermore, the Class-Activation Map also cannot be extracted for ViT-based model. To address this, we propose reducing the number of patches input into the Vision Transformer. With the standard patch size of $16 \times 16$ used by ViT and our chosen image resolution of $112 \times 112$, we generate a $7 \times 7$ grid of patches instead of the original $14 \times 14$. Details of this technique are provided in the **Appendix A.1**.

### 3.4 IMPROVED MODEL DIVERSITY

While lower-resolution synthetic images enhance computational efficiency, they can also limit the model's ability to capture diverse and detailed features, as lower resolutions reduce the space available for representing such diversity.

**Multi-resolution Data Generation.** To overcome this challenge, we propose a multi-resolution generation strategy that synthesizes images at various resolutions, effectively capturing both coarse and fine-grained features. Given a set of resolutions $E$, the synthetic data $\hat{x}$ is generated from each resolution $e \sim E$:

$$\hat{x} = \mathcal{G}_{e \times e, e \in E}(\mathcal{Z}(f_y)), \tag{7}$$

**Embedding Diversity Loss.** Additionally, we introduce embedding diversity techniques to preserve distinct representations within the latent space, ensuring that rich feature representations are maintained even at lower resolutions. These techniques consist of two loss functions, which are used for training the generator $\mathcal{G}$ and the student $\mathcal{S}$, respectively.

---

**Algorithm 1:** K-Gen

---

**Input:** pre-trained teacher $\mathcal{T}_{\theta_{\mathcal{T}}}$, student $\mathcal{S}_{\theta_{\mathcal{S}}}$, generator $\mathcal{G}_{\theta_{\mathcal{G}}}$, text encoder $\mathcal{C}_{\theta_{\mathcal{C}}}$, list of labels $\boldsymbol{y}$ and list of text of these labels $Y_{\boldsymbol{y}}$;

1  Initializing $\mathcal{P} = \{\}, \mathcal{M} = \{\}$;
2  Store all embeddings $\boldsymbol{f_y} = \mathcal{C}(Y_{\boldsymbol{y}})$ into $\mathcal{P}$;
3  **for** $\mathcal{E}$ *epochs* **do**
4     **for** $I$ *iterations* **do**
5         Randomly reinitializing noisy layers $\mathcal{Z}_{\theta_{\mathcal{Z}}}$ and pseudo-label $\hat{\boldsymbol{y}}$ for each iteration;
6         Query $\boldsymbol{f_{\hat{y}}} \sim \mathcal{P}$;
7         **for** $g$ *steps* **do**
8            Sampling $\hat{\boldsymbol{x}} = \mathcal{G}_{e \times e, e \in E}(\mathcal{Z}(\boldsymbol{f_y}))$ and update $\theta_{\mathcal{G}}, \theta_{\mathcal{Z}}$ by minimizing $\mathcal{L}_{\mathcal{G}}$ Eq. 10;
9         $\mathcal{M} \leftarrow \mathcal{M} \cup \hat{\boldsymbol{x}}$;
10    **for** $S$ *iterations* **do**
11        Sampling $\hat{\boldsymbol{x}} \sim \mathcal{M}$ and update $\theta_{\mathcal{S}}$ by minimizing $\mathcal{L}_{\mathcal{S}}$ (Eq. 8);

---

In the student training phase, given a pool of synthetic data $\hat{\boldsymbol{x}} \sim \mathcal{M}$, the student network $\mathcal{S}$ is trained using the following loss function:

$$\mathcal{L}_{\mathcal{S}} = \mathcal{L}_{\text{KL}} + \alpha_{ed}\mathcal{L}_{ed}, \tag{8}$$

$$\mathcal{L}_{ed} = \max\{0, \text{MSE}(\hat{\boldsymbol{f}}_{\mathcal{S}}, \boldsymbol{f_y}) - r_i\}, \tag{9}$$

where $\alpha_{ed}$ is a scaling factor, $\mathcal{L}_{kl}$ is computed by Eq. 2, $\hat{\boldsymbol{f}}_{\mathcal{S}}$ is the latent embedding of $\hat{\boldsymbol{x}}$ in the student model $\mathcal{S}$, and $\boldsymbol{f_y}$ is the class-specific embedding representative. The purpose of the margin term is to learn embeddings from the synthetic data pool $\mathcal{M}$ that are close to the class representative embedding $\boldsymbol{f_y}$ of the original data. Inspired by Tran et al. (2024a), we use the margin loss to encourage $\hat{f}_{\mathcal{S}}$ to stay within an inner radius $r_i$, while preserving its intrinsic distance characteristics.

In the generator training phase, on the other hand, the generator aims to produce a new batch of synthetic data that is positioned far from the class embedding $\boldsymbol{f_y}$. Similar to $\mathcal{L}_{kr}$, we apply a margin loss to ensure that the embedding of $\hat{\boldsymbol{x}}$ in the teacher model $\mathcal{T}$ does not deviate excessively from the desired distribution.

$$\mathcal{L}_{\mathcal{G}} = \alpha_{ce}\mathcal{L}_{ce} + \alpha_{adv}\mathcal{L}_{adv} + \alpha_{bn}\mathcal{L}_{bn} + \alpha_{kr}\mathcal{L}_{kr} + \alpha_{aed}\mathcal{L}_{aed} \tag{10}$$

$$\mathcal{L}_{aed} = \max\{0, r_o - \text{MSE}(\hat{\boldsymbol{f}}_{\mathcal{S}}, \boldsymbol{f_y})\} \tag{11}$$

where $r_o > r_i$ represents the outer radius, and $\alpha$ are scaling parameters.

We now explain how the cooperation between the generator and student in the *embedding in-out game*, achieved by minimizing $\mathcal{L}_{\mathcal{S}}$ and $\mathcal{L}_{\mathcal{G}}$, promotes embedding diversity. Specifically, by minimizing $\mathcal{L}_{ed}$ during student training, the model learns to keep the latent embeddings of all previous data within an inner radius around $\boldsymbol{f_y}$, positioning them closer to $\boldsymbol{f_y}$ (Figure 2 (b)). In contrast, $\mathcal{L}_{aed}$ guides the generator $\mathcal{G}$ to produce new data with latent embeddings that are distant from $\boldsymbol{f_y}$ (Figure 2 (c)). This setup encourages the new data to differ from the old data in latent space, thereby enhancing the diversity of the latent embeddings.

**Choosing Class Representative Embedding $\boldsymbol{f_y}$.** The embedding $\boldsymbol{f_y}$ plays a crucial role in promoting embedding diversity, and we consider two options for selecting $\boldsymbol{f_y}$. First, since we use the generator from NAYER Tran et al. (2024b) as our baseline, we propose using the label text embedding as $\boldsymbol{f_y}$. Second, when the label text embedding is unavailable, we use the mean of the embeddings in $\mathcal{T}$ from the first batch as $\boldsymbol{f_y}$. Both options serve as class representative embeddings. We conducted an ablation study Appendix G showing that both methods are comparable, with the label text embedding yielding slightly better performance.

### 3.5 OVERALL ARCHITECTURE

The overall architecture of K-Gen is shown in Figure 2, and the pseudo code can be found in Algorithm 1. First, K-Gen embeds all label text using either via text encoder or as the mean of T. Then, our method undergoes training for $\mathcal{E}$ epochs. Each epoch consists of two distinct phases:

(i) The first phase involves training the generator. In each iteration $I$, as described in Algorithm 1, the noisy layer $\mathcal{Z}$ is reinitialized (line 5) before being used to learn the label text embedding

Table 1: Distillation results of our K-Gen (multi-resolution) and K-Gen-S (single-resolution) are compared with SOTA DFKD methods—NAYER Tran et al. (2024b), Fast100 Fang et al. (2022) (100 generation steps), and DeepInv Yin et al. (2020)—across datasets (Imagenette, Imagewoof, ImageNet1k) at various data memory ratios. Evaluations cover two common distillation pairs: ResNet50 to MobileNetV2 and ResNet34 to ResNet18. Bold and underlined numbers denote the highest and second-highest accuracies, respectively. Results report the mean accuracy over 3 runs.

| Dataset | Imagenetee | | | | | | | |
|---|---|---|---|---|---|---|---|---|
| Teacher - Student | ResNet50 (92.86) - MobileNetV2 (90.42) | | | | ResNet34 (94.06) - ResNet18 (93.53) | | | |
| Data Memory Ratio | 1% | 5% | 10% | 20% | 1% | 5% | 10% | 20% |
| DeepInv Yin et al. (2020) | 6.71 (4.8h) | 26.02 (6.1h) | 35.31 (8.7h) | 47.02 (13.6h) | 6.03 (3.2h) | 25.08 (4.9h) | 34.04 (6.4h) | 44.65 (9.5h) |
| Fast100 Fang et al. (2022) | 8.92 (0.5h) | 29.18 (0.5h) | 39.12 (0.8h) | 51.43 (1.4h) | 8.51 (0.3h) | 28.32 (0.5h) | 38.25 (0.6h) | 49.11 (1.0h) |
| NAYER Tran et al. (2024b) | 9.54 (0.5h) | 31.28 (0.5h) | 42.24 (0.8h) | 54.26 (1.4h) | 9.35 (0.3h) | 32.17 (0.5h) | 42.57 (0.6h) | 52.72 (1.0h) |
| K-Gen-S (Ours) | 35.32 (0.5h) | 80.11 (0.5h) | 87.21 (0.8h) | 88.53 (1.4h) | 34.52 (0.3h) | 80.32 (0.5h) | 86.67 (0.6h) | 88.25 (1.0h) |
| K-Gen (Ours) | 36.16 (0.8h) | 81.21 (0.8h) | 88.12 (1.2h) | 89.21 (2.1h) | 35.21 (0.5h) | 82.21 (0.8h) | 87.21 (1.1h) | 88.72 (1.5h) |
| Dataset | Imagewoof | | | | | | | |
| Teacher - Student | ResNet50 (86.84) - MobileNetV2 (82.69) | | | | ResNet34 (83.02) - ResNet18 (82.59) | | | |
| Data Memory Ratio | 1% | 5% | 10% | 20% | 1% | 5% | 10% | 20% |
| DeepInv Yin et al. (2020) | 3.68 (2.7h) | 13.26 (5.4h) | 21.34 (7.9h) | 36.01 (14.9h) | 3.42 (2.8h) | 12.62 (5.1h) | 20.97 (7.8h) | 32.42 (10.8h) |
| Fast100 Fang et al. (2022) | 5.42 (0.3h) | 15.11 (0.5h) | 23.45 (0.8h) | 38.92 (1.4h) | 5.21 (0.3h) | 14.24 (0.5h) | 23.54 (0.8h) | 35.72 (1.1h) |
| NAYER Tran et al. (2024b) | 6.99 (0.3h) | 16.72 (0.5h) | 27.43 (0.8h) | 40.21 (1.4h) | 6.72 (0.3h) | 15.62 (0.5h) | 25.27 (0.8h) | 38.25 (1.1h) |
| K-Gen-S (Ours) | 21.25 (0.3h) | 36.24 (0.5h) | 71.42 (0.8h) | 74.53 (1.4h) | 20.52 (0.3h) | 36.25 (0.5h) | 59.85 (0.8h) | 73.74 (1.1h) |
| K-Gen (Ours) | 22.43 (0.5h) | 37.51 (0.8h) | 72.11 (1.2h) | 75.12 (2.1h) | 21.12 (0.5h) | 37.31 (0.8h) | 60.04 (1.2h) | 74.52 (1.5h) |
| Dataset | ImageNet1k | | | | | | | |
| Teacher - Student | ResNet50 (80.86) - MobileNetV2 (71.88) | | | | ResNet34 (73.31) - ResNet18 (69.76) | | | |
| Data Memory Ratio | 1% | 5% | 10% | 20% | 1% | 5% | 10% | 20% |
| DeepInv Yin et al. (2020) | 3.15 (61.2h) | 14.07 (226.3h) | 19.01 (385.0h) | 22.17 (642.7h) | 1.84 (49.6h) | 13.06 (183.1h) | 17.41 (308.3h) | 23.03 (517.9h) |
| Fast100 Fang et al. (2022) | 4.78 (6.3h) | 16.58 (23.5h) | 22.12 (39.2h) | 25.25 (65.6h) | 3.63 (5.1h) | 15.52 (18.8h) | 20.12 (31.4h) | 25.96 (52.5h) |
| NAYER Tran et al. (2024b) | 6.32 (6.3h) | 19.78 (23.5h) | 25.43 (39.2h) | 28.12 (65.6h) | 5.81 (5.1h) | 18.86 (18.8h) | 23.98 (31.4h) | 28.11 (52.5h) |
| K-Gen-S (Ours) | 22.41 (6.3h) | 40.63 (23.5h) | 46.25 (39.2h) | 53.24 (65.6h) | 22.32 (5.1h) | 40.82 (18.8h) | 45.95 (31.4h) | 53.96 (52.5h) |
| K-Gen (Ours) | 24.25 (9.3h) | 42.24 (30.1h) | 47.12 (58.5h) | 54.41 (80.4h) | 24.16 (7.5h) | 42.84 (24.1h) | 47.13 (46.8h) | 54.98 (64.3h) |

$f_y$. The generator and noisy layer are then trained over $g$ steps using Eq. 10 to optimize their performance (line 8).

(ii) The second phase involves training the student network. To mitigate the *risk of forgetting*—which arises in prior DFKD methods like MAD and KAKR that generate, use, and discard synthetic data in each iteration—all generated samples are stored in the memory module $\mathcal{M}$ (line 9), following the strategy proposed in Fang et al. (2022). The student model is then trained using Eq. 8 over $S$ iterations, utilizing samples from $\mathcal{M}$ (lines 10 and 11).

## 3.6 DATA MEMORY RATIO AND COMPARISON FAIRNESS IN DFKD

Training on high-resolution datasets like ImageNet is computationally intensive, particularly with synthetic data generation Yin et al. (2020); Tran et al. (2024b). To manage this, we cap the amount of synthetic data used to train the student model Liu et al. (2024), following practices in Continual Learning Li et al. (2023a;b) and Federated Learning Tran et al. (2024a); Zhu et al. (2021). We evaluate various data memory ratios on ImageNet1k and its subsets; for instance, a 10% ratio yields 100k samples over 1 million training samples at $224 \times 224$ resolution.

**Lower-Resolution Images for Efficiency.** We propose generating lower-resolution images (e.g., $112 \times 112$, $96 \times 96$), which reduces storage and computation. For example, one $224 \times 224$ image is equivalent to four $112 \times 112$ or five $96 \times 96$ images in terms of resource usage. This allows K-Gen to generate more samples, for example, 40k $112 \times 112$ images in a 10% ratio, without increasing memory or training time.

**Single vs. Multi-Resolution Variants.** K-Gen can use mixed resolutions (e.g., 25k $96 \times 96$ + 20k $112 \times 112$) at the same compute cost as 40k $112 \times 112$ images. However, due to PyTorch inefficiencies, multi-resolution training is slower. Thus, we report both K-Gen (multi-resolution) and K-Gen-S (single-resolution) results. Further details and ablation studies are provided in Appendix.

**Is Using More Labels and Lower-Resolution Images in DFKD Fair?** We argue that utilizing a larger number of lower-resolution images and labels, while keeping the total Data Memory Ratio constant, is entirely fair. For example, using 40k synthetic images at $112 \times 112$ resolution instead of 10k images at $224 \times 224$ maintains equivalent memory usage and computational cost. This is because the image generation process operates at the pixel level, and generating four $112 \times 112$ images involves similar time, training effort, and memory consumption as generating one $224 \times 224$ image. Furthermore, **all data and labels are synthetically generated by our model**, without requiring any external data collection or manual annotation. The increase in the number of labels does not translate to additional supervision or unfair advantage. Indeed, several prior works have employed significantly more labels—up to ten times as many—without such concerns being raised Yu et al.

Table 2: The distillation results for the CIFAR10, CIFAR100 and TinyImageNet datasets compare various methods, following the setup of Tran et al. (2024b). The table presents the accuracy achieved by different student models with various architectures, such as ResNet (R) He et al. (2016), VGG (V) Simonyan & Zisserman (2014), and WideResNet (W) Zagoruyko & Komodakis (2016). The results from compared methods are collected at Tran et al. (2024b).

| Method | CIFAR10 | | | | | CIFAR100 | | | | | TinyImageNet |
|---|---|---|---|---|---|---|---|---|---|---|---|
| | R34 R18 | W402 W162 | W402 W161 | W402 W401 | V11 R18 | R34 R18 | W402 W162 | W402 W161 | W402 W401 | V11 R18 | R34 R18 |
| Teacher | 95.70 | 94.87 | 94.87 | 94.87 | 92.25 | 77.94 | 77.83 | 75.83 | 75.83 | 71.32 | 66.44 |
| Student | 95.20 | 93.95 | 91.12 | 93.94 | 95.20 | 77.10 | 73.56 | 65.31 | 72.19 | 77.10 | 64.87 |
| DeepInv Yin et al. (2020) | 93.26 | 89.72 | 83.04 | 86.85 | 90.36 | 61.32 | 61.34 | 53.77 | 68.58 | 54.13 | - |
| DFQ Choi et al. (2020) | 94.61 | 92.01 | 86.14 | 91.69 | 90.84 | 77.01 | 64.79 | 51.27 | 54.43 | 66.21 | - |
| ZSKT Micaelli & Storkey (2019) | 93.32 | 89.66 | 83.74 | 86.07 | 89.46 | 67.74 | 54.59 | 36.60 | 53.60 | 54.31 | - |
| CMI Fang et al. (2021) | 94.84 | 92.52 | 90.01 | 92.78 | 91.13 | 77.04 | 68.75 | 57.91 | 68.88 | 70.56 | 64.01 |
| PREKD Binici et al. (2022b) | 93.41 | - | - | - | - | 76.93 | - | - | - | - | 49.94 |
| MBDFKD Binici et al. (2022c) | 93.03 | - | - | - | - | 76.14 | - | - | - | - | 47.96 |
| FM Fang et al. (2022) | 94.05 | 92.45 | 89.29 | 92.51 | 90.53 | 74.34 | 65.12 | 54.02 | 63.91 | 67.44 | - |
| MAD Do et al. (2022) | 94.90 | 92.64 | - | - | - | 77.31 | 64.05 | - | - | - | 62.32 |
| KAKR_MB Patel et al. (2023) | 93.73 | - | - | - | - | 77.11 | - | - | - | - | 47.96 |
| KAKR_GR Patel et al. (2023) | 94.02 | - | - | - | - | 77.21 | - | - | - | - | 49.88 |
| SpaceshipNet Yu et al. (2023) | 95.39 | 93.25 | 90.38 | 93.56 | 92.27 | 77.41 | 69.95 | 58.06 | 68.78 | 71.41 | 64.04 |
| **NAYER Tran et al. (2024b)** | 95.21 | 94.07 | 91.94 | 94.15 | 92.37 | 77.54 | 71.72 | 62.23 | 71.80 | 71.75 | 64.17 |
| K-Gen-S | 95.36 | 94.35 | 92.27 | 94.37 | 93.02 | 77.64 | 72.21 | 62.87 | 72.01 | 71.94 | 64.41 |
| **K-Gen** | **95.41** | **94.39** | **92.32** | **94.44** | **93.20** | **77.78** | **72.31** | **62.92** | **72.13** | **72.11** | **64.54** |

(2023); Patel et al. (2023). Hence, we affirm that our comparison adheres to fair and consistent evaluation standards.

# 4 EXPERIMENTS

## 4.1 EXPERIMENTAL SETTINGS

For high-resolution datasets, we evaluated our method using two commonly used backbone pairs: ResNet34/ResNet18 He et al. (2016) and ResNet50/MobileNetV2 Sandler et al. (2018), on ImageNet1k Deng et al. (2009), which comprises 1,000 object categories and over 1.2 million labeled training images. We also included its subsets, ImageNette and ImageWoof Howard (2019b), each consisting of 10 specific subclasses. For low-resolution datasets, we conducted experiments using ResNet, VGG Simonyan & Zisserman (2014), and WideResNet (WRN) Zagoruyko & Komodakis (2016) across CIFAR-10, CIFAR-100 Krizhevsky et al. (2009), and Tiny ImageNet Le & Yang (2015). Additional details on all datasets used in this paper, the architectures, parameter settings, parameter sensitivity and further analysis can be found in the **Appendix C**.

## 4.2 RESULTS AND ANALYSIS

**Comparison on High-resolution Datasets ($>$ 100k Pixels).** Table 1 presents the distillation results across multiple datasets, including Imagenette, Imagewoof, and ImageNet1k ($3 \times 224 \times 224$ pixels), comparing the performance of K-Gen-S and K-Gen with existing methods such as DeepInv Yin et al. (2020), Fast100 Fang et al. (2022), and NAYER Tran et al. (2024b) at varying data memory ratios. Overall, both K-Gen and K-Gen-S consistently achieve superior performance, with at least a two-digit improvement in all comparison cases, while still maintaining low training time. This performance gain can be attributed to the use of multi-resolution strategies and key region generation techniques, which are particularly beneficial for high-resolution datasets like ImageNet1k and its subsets. These results clearly demonstrate the effectiveness of our proposed approach.

**Comparison on Low-resolution Datasets ($\approx$ 1k Pixels).** We also conducted experiments on low-resolution datasets such as CIFAR-10, CIFAR-100, and TinyImageNet, in Table 2. The results demonstrate the strong performance of both K-Gen and K-Gen-S compared to existing methods. However, the performance gains in these tasks are less pronounced than those observed on high-resolution datasets. This can be attributed to two main factors: (1) the current accuracy on these low-resolution datasets is already close to the upper bound defined by the teacher model, and (2) our proposed techniques are primarily designed to enhance distillation performance on high-resolution data, making them less effective for lower-resolution datasets like CIFAR-10.

**Comparison on Mega-resolution Datasets ($>$ 1M Pixels).** To further assess the generalizability of our method, we evaluate K-Gen on two additional mega-resolution datasets: Traffic Sign

Table 3: Additional results on mega-resolution datasets. K-Gen outperforms NAYER across all settings under varying data memory ratios.

| Method | Traffic Sign Recognition (1200 × 1600 pixels) | | Megapixel MNIST (1500 × 1500 pixels) | |
|---|---|---|---|---|
| | 10% | 20% | 10% | 20% |
| Teacher/Student | CNN (84.1) / CNN (84.1) | | CNN (91.9) / CNN (91.9) | |
| Fast100 | 41.67 | 54.12 | 49.12 | 58.21 |
| NAYER | 48.23 | 57.32 | 52.47 | 63.91 |
| K-Gen | **72.43** | **77.56** | **78.24** | **84.12** |

Table 4: Ablation studies for all combinations of the proposed components with the Data Memory Ratio at 5%.

| Method | Imagenette | Imagewoof | ImageNet1k |
|---|---|---|---|
| NAYER | 32.17 | 15.62 | 18.86 |
| +SRG | 41.13 | 19.71 | 23.87 |
| +SRG+KR | 77.62 | 33.22 | 37.41 |
| +SRG+KR+ED (K-Gen-S) | 80.32 | 36.25 | 40.82 |
| +MRG | 46.25 | 22.14 | 24.95 |
| +MRG+KR | 79.92 | 36.92 | 39.81 |
| +MSG+KR+ED (K-Gen) | **82.21** | **37.31** | **42.84** |

Recognition ($3 \times 1200 \times 1600$ pixels) Katharopoulos & Fleuret (2019) and Megapixel MNIST ($1 \times 1500 \times 1500$ pixels) Katharopoulos & Fleuret (2019). As shown in Table 3, K-Gen consistently outperforms the baseline method NAYER across both datasets and under varying data memory ratios (10% and 20%). On the Traffic Sign Recognition task, K-Gen achieves notable improvements of over 20% in nearly all cases. These results demonstrate that K-Gen scales effectively to complex, mega-resolution visual tasks, opening the door to broader applications of the field in high-resolution vision domains.

### 4.3 FURTHER ANALYSIS

**Ablation Study: Components Analysis.** To better understand the contribution of each component in our proposed method, we conduct a comprehensive ablation study under a consistent Data Memory Ratio of 5%, as shown in Table 4. Starting from the baseline (NAYER), we incrementally add our proposed modules: SRG (Smaller-Resolution Generation) which only generate smaller resolution image with KR loss, KR (Key Region Data Generation, Section 3.2), ED (Embedding Diversity Loss, Section 3.4), and MSG (Multi-Resolution Data Generation, Section 3.4). The experiment demonstrate that: (1) each component individually enhances performance across all datasets. (2) the Key Region module plays a crucial role, significantly boosting performance—for example, from 41.13% (+SRG) to 77.62% (+SRG+KR), and from 46.25% (+MRG) to 79.92% (+MRG+KR).

**Comparison for ViT Model.** To demonstrate the effectiveness of our approach on ViT-based models, we conducted experiments comparing our K-Gen with NAYER, using DeiT-B (Teacher) and DeiT-Tiny (Student) on ImageNet-1K. As shown in Table 5, K-Gen outperforms the original NAYER training, achieving double-digit improvements.

Table 5: Performance Comparison Our K-Gen and NAYER in DeiT-B (Teacher) and DeiT-Tiny (Student) on ImageNet-1K.

| Data Memory Ratio | 1% | | 5% | |
|---|---|---|---|---|
| Metric (Accuracy) | Top 1 (%) | Top 5 (%) | Top 1 (%) | Top 5 (%) |
| NAYER | 4.52 | 19.45 | 16.24 | 43.24 |
| K-Gen | 15.24 | 36.52 | 28.24 | 60.24 |

## 5 CONCLUSION

In this paper, we propose K-Gen, a novel approach to overcome the limitations of traditional DFKD methods on high-resolution datasets. K-Gen synthesizes lower-resolution images guided by Class Activation Maps to preserve class-specific features, reducing noise and computational cost, particularly on large-scale datasets like ImageNet1K. Through multi-resolution synthesis and embedding diversity, K-Gen enriches learned representations and boosts student model performance. Experiments show that K-Gen achieves state-of-the-art results with double-digit gains on ImageNet1K and remains effective on mega-resolution datasets (over one million pixels), enabling broader applications in vision field.

**Limitation and Future work:** Our paper employs a customized version of the classic CAM, designed to facilitate backpropagation in obtaining the activation matrix. This approach opens the door to exploring other techniques, such as Grad-CAM Selvaraju et al. (2017) or attention-based scores Leem & Seo (2024), to further enhance the task. Additionally, optimizing multi-resolution techniques for faster processing times presents another promising direction for improvement.

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

## A  APPENDIX

### A.1  LOWER-RESOLUTION IMAGE FOR VISION TRANSFORMER

A key challenge in our approach is training the student model with lower-resolution images, which are then tested on full-resolution data. This is particularly challenging for patch-based models, such as Vision Transformer (ViT) and its variants Dosovitskiy (2020); Touvron et al. (2021), that do not rely on CNN architectures. Furthermore, the Class-Activation Map also cannot be extracted for ViT-based model. To address this, we propose reducing the number of patches input into the Vision Transformer. With the standard patch size of $16 \times 16$ used by ViT and our chosen image resolution of $112 \times 112$, we generate a $7 \times 7$ grid of patches instead of the original $14 \times 14$. By focusing on the center position embedding, our method, as shown in Table 5, outperforms the original NAYER training, achieving improvements of over two percentage points. Details of this technique are provided in the Appendix A.1.

To illustrate the patch-reduction strategy mathematically, consider the input image resolution $H \times W$. The Vision Transformer (ViT) splits the image into patches of size $P \times P$, resulting in a grid of $\frac{H}{P} \times \frac{W}{P}$ patches. For the standard ViT, with $P = 16$, and full-resolution images $H = 224$ and $W = 224$, the number of patches is:

$$N_{\text{patches}} = \frac{H}{P} \cdot \frac{W}{P} = \frac{224}{16} \cdot \frac{224}{16} = 14 \cdot 14 = 196. \tag{12}$$

For our approach, we reduce the resolution to $H = 112$ and $W = 112$, while maintaining $P = 16$. This results in:

$$N_{\text{patches}} = \frac{H}{P} \cdot \frac{W}{P} = \frac{112}{16} \cdot \frac{112}{16} = 7 \cdot 7 = 49. \tag{13}$$

**Position Embedding.** Let the index matrix $\mathcal{I}$ be a $10 \times 10$ grid, where both row and column values range from 2 to 12:

$$\mathcal{I} = \{(r, c) \mid 2 \le r \le 12, 2 \le c \le 12\}.$$

We randomly select the center index $p_{\text{center}} = (p^r_{\text{center}}, p^c_{\text{center}})$ from this grid with a bias toward the center, particularly around indices $7$ and $8$ for both rows and columns. The probability of selecting the center index $p_{\text{center}}$ is given by:

$$P(p_{\text{center}}) \propto \frac{1}{1 + \lambda \cdot (|p^r_{\text{center}} - 7|^2 + |p^c_{\text{center}} - 7|^2)},$$

where:

- $(p^r_{\text{center}}, p^c_{\text{center}})$ are the indices in the grid,
- $\lambda$ is a parameter that controls the steepness of the decay, influencing how strongly the selection is biased toward the center,
- $|p^r_{\text{center}} - 7|^2 + |p^c_{\text{center}} - 7|^2$ represents the squared Euclidean distance from the center index $(7, 7)$.

This formulation ensures that the selection probability decreases as the distance from the center increases, making the center indices $(7, 7)$ and $(8, 8)$ more likely to be chosen.

**Patch Index Mapping.** After selecting the center index $p_{\text{center}} = (r, c)$, the synthetic image patches are indexed relative to $p_{\text{center}}$. Let $p_i$ represent the index of the patch. The patch indices $p_i$ are determined by an offset from $p_{\text{center}}$. For a patch size of $P \times P$, the patch index $p_i$ is defined as:

$$p_i = (p^r_{\text{center}} + \Delta r, p^c_{\text{center}} + \Delta c),$$

where $\Delta r, \Delta c \in \{-P, 0, P\}$ and are the offsets applied to the center index $p_{\text{center}}$. This allows the selection of patches in a surrounding area around the center index $p_{\text{center}}$. This approach ensures that patch indices closer to the center are more likely to be selected, with the probability decreasing as the distance from the center increases.

**Attention Map as a Replacement for CAM:** In ViT-based models, the attention map with respect to the [CLS] token can serve as a substitute for Class Activation Maps (CAM), which typically do not function effectively in ViT architectures. In fact, they share a similar ability to highlight class-relevant regions, enabling effective visual explanations.

## B  DATASETS

Table 6 summarizes all the datasets used in our paper, including three low-resolution, three high-resolution, and two mega-resolution datasets.

Table 6: Overview of benchmark datasets categorized by resolution.

| Dataset | Image Size | #Classes | #Train Images | #Test Images |
|---|---|---|---|---|
| **Low-Resolution Datasets** | | | | |
| CIFAR-10 Krizhevsky et al. (2009) | 32×32×3 | 10 | 50,000 | 10,000 |
| CIFAR-100 Krizhevsky et al. (2009) | 32×32×3 | 100 | 50,000 | 10,000 |
| Tiny ImageNet Le & Yang (2015) | 64×64×3 | 200 | 100,000 | 10,000 |
| **High-Resolution Datasets** | | | | |
| ImageNette Howard (2019a) | 224×224×3 | 10 | 9,469 | 3,925 |
| ImageWoof Howard (2019b) | 224×224×3 | 10 | 9,902 | 3,926 |
| ImageNet-1k Deng et al. (2009) | 224×224×3 | 1,000 | 1,281,167 | 50,000 |
| **Mega-Resolution Datasets** | | | | |
| Traffic Sign Recognition Katharopoulos & Fleuret (2019) | 1200×1600×3 | 4 | 747 | 684 |
| Megapixel MNIST (Max Digit Recognition) Katharopoulos & Fleuret (2019) | 1500×1500×1 | 10 | 5000 | 1000 |

## C  TRAINING DETAILS

In this section, we provide the details of model training for our methods, including Teacher Training, Generator, and Student Training.

## C.1 TEACHER MODEL TRAINING DETAILS

In this work, we utilized the pretrained ResNet-50 and ResNet-34 models from PyTorch, trained on ImageNet1k, and trained them from scratch on the ImageNette and ImageWoof datasets. For CIFAR-10/CIFAR-100, we employed pretrained ResNet-34 and WideResNet-40-2 teacher models from Fang et al. (2022); Tran et al. (2024b). The teacher models were trained using the SGD optimizer with an initial learning rate of 0.1, momentum of 0.9, and weight decay of 5e-4, with a batch size of 128 for 200 epochs. The learning rate decay followed a cosine annealing schedule.

## C.2 GENERATOR TRAINING DETAILS

To ensure fair comparisons, we adopt the generator architecture outlined in Fang et al. (2022); Tran et al. (2024b) and the Noisy Layer (`BatchNorm1D, Linear`) as described in Tran et al. (2024b) for all experiments. This architecture has been proven effective in prior work and provides a solid foundation for evaluating the performance of our model. The generator network is designed to learn rich feature representations while maintaining computational efficiency. The details of the generator architecture, including layer specifications and output sizes, are provided in Table 7. Additionally, we use the Adam optimizer with a learning rate of 4e-3 to optimize the generator, ensuring stable convergence during training.

Table 7: Architecture of the Generator Network ($\mathcal{G}$), detailing the sequence of operations and layer sizes from input to output. The network includes linear transformations, spectral normalization in convolution layers, batch normalization, leaky ReLU activations, upsampling, and a sigmoid activation for the output. Output dimensions at each layer are shown in relation to the input height (h) and width (w), with intermediate feature maps gradually upscaled to the final $3 \times h \times w$ generated image.

| Output | Size Layers |
|---|---|
| 1000 | Input |
| $128 \times h/4 \times w/4$ | Linear |
| $128 \times h/4 \times w/4$ | BatchNorm1D |
| $128 \times h/4 \times w/4$ | Reshape |
| $128 \times h/2 \times w/2$ | SpectralNorm (Conv (3 × 3)) |
| $128 \times h/2 \times w/2$ | BatchNorm2D |
| $128 \times h/2 \times w/2$ | LeakyReLU |
| $128 \times h/2 \times w/2$ | UpSample (2×) |
| $64 \times h \times w$ | SpectralNorm (Conv (3 × 3)) |
| $64 \times h \times w$ | BatchNorm2D |
| $64 \times h \times w$ | LeakyReLU |
| $64 \times h \times w$ | UpSample (2×) |
| $3 \times h \times w$ | SpectralNorm (Conv (3 × 3)) |
| $3 \times h \times w$ | Sigmoid |
| $3 \times h \times w$ | BatchNorm2D |

## C.3 STUDENT MODEL TRAINING DETAILS

In all experiments, we adopt a consistent approach for training the student model. The batch size is set to match the Synthetic Batch Size, and the AdamW optimizer is used with a momentum of 0.9 and an initial learning rate of 1e-3. To further optimize training, a lambda scheduler is employed to adjust the learning rate dynamically throughout the training process.

## C.4 OTHER SETTINGS

We trained the model for $\mathcal{E}$ epochs, incorporating a warm-up phase during the first 10% of $\mathcal{E}$, as outlined in the settings defined in Fang et al. (2022); Tran et al. (2024b). This warm-up phase gradually increases the learning rate to stabilize training early on. Additionally, the model was trained with the specified batch size and other hyperparameters, which were carefully selected to ensure optimal performance. Further details regarding these parameters, including their values and any adjustments made during the training process, are provided in Table 8.

Table 8: The hyperparameters used in our methods across five different datasets are detailed below. **Image Resolution** and **Synthetic Batch Size** refer to the resolution and batch size of synthetic images generated by our methods. Notably, in the case of K-Gen, two different resolutions are used, and their batch sizes are adjusted based on their scales. Other key parameters include: $S$, the number of training steps for optimizing the student model, scaled based on the data memory ratio ($d_r$); $I$, the number of times a batch of images is generated per epoch; and $g$, the training steps for optimizing the generators. Additionally, the following hyperparameters were fixed for all experiments: $\alpha_{ce} = 0.5$, $\alpha_{bn} = 10$, $\alpha_{adv} = 1.3$ (as in Tran et al. (2024b)). Furthermore, in our paper, we propose the following parameters, which are also fixed for all experiments (their parameter sensitivity analysis can be found in Section D.7): $\alpha_{kr} = 0.1$ (for Key Region Loss); $\alpha_{ed} = 10$, $\alpha_{aed} = 5$, $r_i = 0.015$, and $r_o = 0.03$ (for Embedding Diversity).

| | Method | Image Resolution | Synthetic Batch Size | $S$ | $I$ | $g$ | Epoch $\mathcal{E}$ |
|---|---|---|---|---|---|---|---|
| ImageNettee/ImageWoof | K-Gen-S | $96 \times 96$ | 100 | $50 \times d_r$ | 5 | 100 | 100 |
| | K-Gen | $[96 \times 96, 112 \times 112]$ | $[50, 40]$ | | | | |
| ImageNet1k | K-Gen-S | $112 \times 112$ | 200 | $200 \times d_r$ | 20 | 100 | 400 |
| | K-Gen | $[112 \times 112, 128 \times 128]$ | $[200, 150]$ | | | | |
| CIFAR10/CIFAR100 | K-Gen-S | $28 \times 28$ | 260 | $2 \times d_r$ | 20 | 40 | 400 |
| | K-Gen | $[28 \times 28, 32 \times 32]$ | $[130, 100]$ | | | | |
| TinyImageNet | K-Gen-S | $32 \times 32$ | 200 | $50 \times d_r$ | 5 | 100 | 100 |
| | K-Gen | $[32 \times 32, 48 \times 48]$ | $[200, 100]$ | | | | |
| Traffic Sign Recognition/Megapixel MNIST | K-Gen-S | $112 \times 112$ | 260 | $50 \times d_r$ | 5 | 100 | 100 |
| | K-Gen | $[112 \times 112, 128 \times 128]$ | $[200, 150]$ | | | | |

# D   FURTHER ABALATION STUDY

## D.1   COMPARASION IN HIGHER DATA MEMORY RATIOS.

To further demonstrate the benefits of our methods, we also conducted experiments on higher data memory ratio settings, as shown in Figure 3a-b. The results indicate that our methods achieve higher accuracy across all ratio settings on both the Imagenette and Imagewoof datasets. Particularly at lower ratios, the difference is significant. For example, at a ratio of 20% on Imagenette, our K-Gen method achieves an accuracy approximately 40% higher than the compared methods. These results demonstrate the effectiveness of our models.

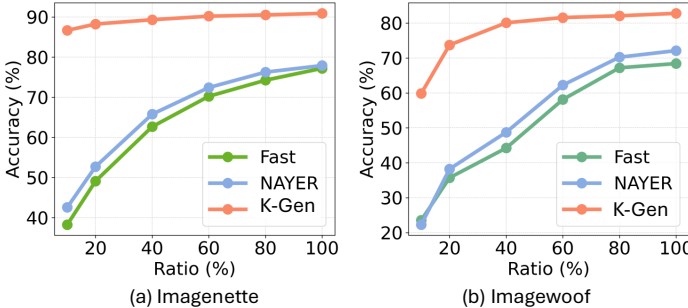

Figure 3: The accuracy at data ratios from 10% to 100% is shown for the teacher (ResNet34) and student (ResNet18) models.

## D.2   TRAINING TIME FOR LOW-RESOLUTION DATASET

As shown in Table 9, while achieving SOTA accuracy, our K-Gen (9.45h) and K-Gen-S (6.84h) also have comparable runtimes to previous methods like NAYER (6.78h) and Fast10 (7.02h), while being significantly faster than DeepInv (31.24h) and CMI (24.01h).

Table 9: Comparing training times in hours using a single NVIDIA A100 for DFKD methods on CIFAR-10 and CIFAR-100 with the teacher/student models WRN40-2/WRN16-2.

| | DeepInv | CMI | DFQ | ZSKT | MAD | SpaceshipNet | Fast10 | NAYER | K-Gen-S | K-Gen |
|---|---|---|---|---|---|---|---|---|---|---|
| CIFAR10 | 89.72 (31.23h) | 92.52 (24.01h) | 92.01 (3.31h) | 89.66 (3.44h) | 92.64 (13.13h) | 93.25 (14.48h) | 92.31 (7.02h) | 94.07 (6.78h) | 94.15 (6.84h) | 94.25 (9.45h) |
| CIFAR100 | 61.34 (31.23h) | 68.75 (24.01h) | 64.79 (3.31h) | 54.59 (3.44h) | 64.05 (26.45h) | 69.95 (29.24h) | 68.25 (7.56h) | 71.72 (7.22h) | 72.12 (7.25h) | 72.32 (9.86h) |
| Avergaing Speed Up | 1.00× | 1.30× | 9.73× | 9.08× | 1.78× | 14.17× | 7.46× | 4.29× | 4.47× | 3.17× |

## D.3   EFFECTIVENESS OF EMBEDDING DIVERSITY LOSS.

Figure 4a shows that K-Gen with Embedding Diversity Loss consistently outperforms without ED at all data memory ratios, especially at lower ratios (1% and 5%), emphasizing ED's crucial role.

Additionally, Figure 4b illustrates that new data typically occupies a distinct region in latent space, enhancing model diversity.

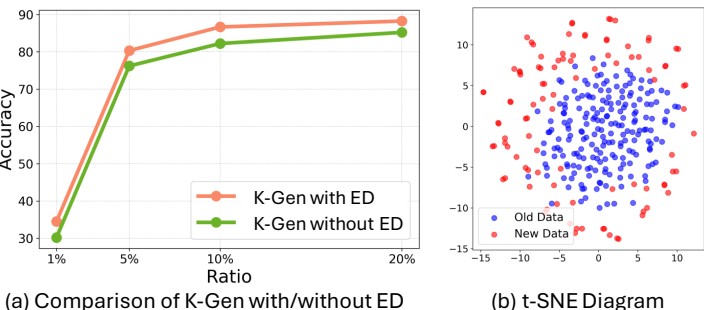

(a) Comparison of K-Gen with/without ED          (b) t-SNE Diagram

Figure 4: (a) Accuracy of our K-Gen method with and without Embedding Diversity (ED) for ResNet34 and ResNet18. (b) t-SNE visualization of the embeddings: synthetic data from the $\mathcal{M}$ pool (blue) and newly generated data (red).

## D.4 EFFECTIVENESS OF LOWER-RESOLUTION.

In Table 10, we present the accuracy of our methods with different image resolutions. The results clearly show that the accuracy of models decreases significantly when the resolution is either too small ($64 \times 64$) or too large ($224 \times 224$), with the highest accuracy achieved at $96 \times 96$. This illustrates the importance of selecting an appropriate resolution for synthetic data, balancing both computational efficiency and model performance.

| Resolution ($R \times R$) | 224 | 192 | 144 | 128 | 112 | 96 | 80 | 64 |
|---|---|---|---|---|---|---|---|---|
| With $\mathcal{L}_{kr}$ | 37.27 | 40.65 | 65.25 | 70.21 | 78.21 | 80.32 | 77.21 | 40.21 |
| Without $\mathcal{L}_{kr}$ | 32.17 | 34.26 | 58.21 | 65.21 | 72.25 | 75.12 | 71.23 | 34.91 |

Table 10: Performance comparison across multiple data resolutions in Imagenette (ResNet34/ResNet18 case) with the same Data Memory Ratio at 5%.

## D.5 EFFECTIVENESS OF KEY REGION DATA GENERATION.

As shown in Table 10, adding the key region loss term, $\mathcal{L}_{kr}$, improves performance, particularly at intermediate resolutions like $128 \times 128$ and $112 \times 112$. At these resolutions, the model achieves 70.21% and 78.21% accuracy, outperforming settings without $\mathcal{L}_{kr}$ by 5-6 percentage points, highlighting its effectiveness, especially at lower resolutions.

## D.6 EFFECTIVENESS OF MULTI-RESOLUTION DATA GENERATION.

Tables 1 and 2 demonstrate that K-Gen, using multi-scale data generation, outperforms other distillation methods in both accuracy and efficiency across various datasets. For instance, on CIFAR10, K-Gen achieves 94.51% accuracy, surpassing NAYER and SSD-KD. Similarly, on CIFAR100, K-Gen reaches 75.21%, outperforming K-Gen-S and NAYER, while also delivering superior performance on Imagenette, showcasing its robustness.

## D.7 PARAMETER SENSITIVITY ANALYSIS

All experiments in this section were conducted in ImageNette and ImageWoof (Resnet34/Resnet18) with ratios at 5% and 10%.

**Parameter $\alpha_{kr}$.** In Table 11, we compare the impact of different scale factors on Key Region Loss. The results show that our methods perform well, achieving higher accuracy with smaller scaling factors, peaking at a scale factor of 0.1. This can be attributed to the fact that the value of the key region generation function is high due to direct subtract function, and a smaller scale factor is more effective for normalizing it.

**Parameters $\alpha_{ed}$ and $\alpha_{aed}$.** Tables 12 and 13 compare the performance of different values of $\alpha_{\text{ed}}$ and $\alpha_{\text{aed}}$ on the ImageNette and ImageWoof datasets at 5% and 10% data memory ratio.In both

Table 11: Comparison of the impact of various scale factors on Key Region Loss, highlighting the optimal performance achieved with smaller scale factors, peaking at a scale factor of 0.1.

| $\alpha_{\mathrm{kr}}$ | 0.05 | 0.1 | 0.2 | 0.5 | 1 | 2 |
|---|---|---|---|---|---|---|
| ImageNette (5%) | 79.77 | **80.32** | 80.2 | 79.69 | 78.26 | 78.63 |
| ImageNette (10%) | 86.32 | **86.67** | 86.18 | 85.64 | 85.16 | 85.41 |
| ImageWoof (5%) | 36.03 | **36.25** | 35.67 | 35.66 | 35.13 | 35.11 |
| ImageWoof (10%) | 59.83 | **59.85** | 59.75 | 59.75 | 58.17 | 57.92 |

tables, the highest accuracy is typically observed at intermediate values of $\alpha$, with $\alpha_{\mathrm{ed}} = 10$ and $\alpha_{\mathrm{aed}} = 5$ yielding the best results in most cases. his can be attributed to the fact that at these values, the mean squared error (MSE) distance between embeddings is significantly small. For instance, the minimum distance between two label text embeddings is just 0.03, which necessitates a higher scaling factor to amplify the impact of this term.

Table 12: Performance comparison of different $\alpha_{\mathrm{ed}}$ values on the ImageNette and ImageWoof datasets at 5% and 10% sampling rates. The highest accuracy is achieved at $\alpha_{\mathrm{ed}} = 10$, highlighting the importance of balancing the scaling factor to minimize MSE distance between embeddings.

| $\alpha_{\mathrm{ed}}$ | 1 | 2 | 5 | 10 | 20 | 50 |
|---|---|---|---|---|---|---|
| ImageNette (5%) | 80.02 | 79.62 | 79.59 | **80.12** | 79.79 | 80.27 |
| ImageNette (10%) | 86.18 | 86.36 | 86.52 | **86.77** | 85.91 | 86.64 |
| ImageWoof (5%) | 35.77 | 35.69 | 36.12 | **36.31** | 35.37 | 36.13 |
| ImageWoof (10%) | 59.52 | 59.54 | 59.82 | **59.91** | 58.60 | 59.70 |

Table 13: Performance comparison of different $\alpha_{\mathrm{aed}}$ values on the ImageNette and ImageWoof datasets at 5% and 10% sampling rates. Peak accuracy is observed at $\alpha_{\mathrm{aed}} = 5$, emphasizing the role of scaling to optimize the MSE distance between embeddings.

| $\alpha_{\mathrm{aed}}$ | 1 | 2 | 5 | 10 | 20 | 50 |
|---|---|---|---|---|---|---|
| ImageNette (5%) | 79.85 | 79.56 | **80.42** | 79.88 | 80.27 | 80.25 |
| ImageNette (10%) | 86.58 | 86.01 | **86.68** | 85.56 | 86.17 | 85.67 |
| ImageWoof (5%) | 35.72 | 35.28 | **36.31** | 35.83 | 35.04 | 35.96 |
| ImageWoof (10%) | 59.35 | 58.88 | **59.88** | 59.54 | 59.47 | 59.52 |

**Inner Radius $r_i$ and Outer Radius $r_o$.** In this approach, we follow the method proposed in Tran et al. (2024a) to determine the most effective radius. Based on this, we found that the minimum distance between two label text embeddings is 0.03. Therefore, we define the inner and outer radii around this value. As shown in Table 14, the pair of 0.015 ($r_i$) for the inner radius and 0.03 ($r_o$) for the outer radius yields the highest accuracy. This demonstrates that half of the minimum distance is optimal for the inner radius of Bounding Loss, similar to Tran et al. (2024a), while the full minimum distance serves as the most effective outer radius.

Table 14: Comparison of different inner ($r_i$) and outer ($r_o$) radius pairs for Bounding Loss and Marging Loss for Embedding Diversity terms. The pair of 0.015 for the inner radius and 0.03 for the outer radius achieves the highest accuracy, demonstrating that half of the minimum distance between embeddings works best for the inner radius, while the full minimum distance is optimal for the outer radius.

| $r_o$ \ $r_i$ | **0.05** | **0.015** | **0.03** | **0.05** | **0.1** |
|---|---|---|---|---|---|
| **0.01** | 76.30 | 80.21 | 80.20 | 80.13 | 76.33 |
| **0.03** | 77.33 | **80.44** | 80.24 | 79.08 | 76.32 |
| **0.1** | 79.17 | 79.07 | 79.16 | 77.46 | 76.42 |
| **0.3** | 78.09 | 78.19 | 78.02 | 77.41 | 76.36 |
| **1** | 76.37 | 76.32 | 76.46 | 76.43 | 76.37 |

## D.8 MIXED RESOLUTION ANALYSIS.

To analyze the robustness of K-Gen under mixed-resolution training, we evaluate its performance across a wide range of resolution combinations on Imagenette and ImageNet1k with a fixed 5% data memory ratio, as reported in Tables 15 and 16. Each row corresponds to the base training resolution, while each column indicates the testing resolution. On Imagenette (Table 15), we observe a consistent increase in accuracy as the training resolution decreases from 224 to 112, with

peak performance (**82.21**%) achieved at $112 \times 112$. This suggests that training on moderately lower resolutions can enhance the generalization capability of K-Gen, particularly in data-scarce settings. Similarly, on ImageNet1k (Table 16), the model achieves its highest accuracy of **42.25%** when trained at $128 \times 128$, demonstrating that K-Gen remains effective even when scaling to more complex and high-resolution datasets. These results highlight K-Gen's adaptability and efficiency in handling resolution variability—an essential trait for practical deployment in resource-constrained or dynamically changing environments.

Table 15: Accuracy of K-Gen with ResNet34/ResNet18 on Imagenette (5% Data Ratio) across various mixed resolutions. Rows represent training resolution; columns represent testing resolution.

| Resolution | 192 | 144 | 128 | 112 | 96 | 80 | 64 |
|---|---|---|---|---|---|---|---|
| 224 | 39.86 | 42.15 | 50.60 | 56.18 | 68.58 | 58.64 | 39.32 |
| 192 | – | 44.99 | 58.16 | 65.17 | 77.55 | 68.95 | 44.52 |
| 144 | – | – | 65.37 | 68.78 | 78.32 | 72.27 | 48.67 |
| 128 | – | – | – | 70.71 | 80.32 | 71.75 | 53.92 |
| 112 | – | – | – | – | **82.21** | 81.67 | 62.42 |
| 96 | – | – | – | – | – | 78.18 | 55.45 |
| 80 | – | – | – | – | – | – | 40.99 |

Table 16: Accuracy of K-Gen with ResNet34/ResNet18 on ImageNet1k (5% Data Memory Ratio) across various mixed resolutions. Rows represent training resolution; columns represent testing resolution.

| Resolution | 192 | 144 | 128 | 112 | 96 | 80 | 64 |
|---|---|---|---|---|---|---|---|
| 224 | 19.25 | 22.80 | 32.16 | 30.76 | 29.27 | 27.76 | 21.67 |
| 192 | – | 23.38 | 37.46 | 35.73 | 34.90 | 32.09 | 27.91 |
| 144 | – | – | 41.11 | 38.35 | 37.73 | 34.86 | 33.99 |
| 128 | – | – | – | **42.25** | 40.75 | 37.93 | 35.60 |
| 112 | – | – | – | – | 38.19 | 36.88 | 33.73 |
| 96 | – | – | – | – | – | 34.76 | 28.65 |
| 80 | – | – | – | – | – | – | 22.40 |

# E    OTHER RESULTS

## E.1    RESULTS ON SEMANTIC SEGMENTATION.

We further examine the generalization capability of K-Gen by conducting experiments on the NYUv2 dataset. Unlike prior DFKD methods that generate data at a resolution of $256 \times 256$, K-Gen operates at a lower resolution of $128 \times 128$. Despite this, it consistently delivers better semantic segmentation performance, as shown in Table 17.

Although segmentation is widely used to evaluate DFKD methods, its effectiveness is often limited by poor-quality synthetic data and the difficulty of generating reliable labels. Nevertheless, K-Gen achieves state-of-the-art results, demonstrating its ability to generalize well even under such challenging conditions.

Table 17: Performance comparison of K-Gen with existing DFKD methods on the NYUv2 dataset.

| Method | DFAD | DAFL | Fast | NAYER | K-Gen |
|---|---|---|---|---|---|
| Synthetic Time | 6.0h | 3.99h | 0.82h | 0.82h | 0.82h |
| mIoU | 0.364 | 0.105 | 3.66 | 3.85 | **4.01** |

## E.2    ERROR BAR

Table 18 show that our method consistently achieves higher accuracy across three runs with only minor standard deviation, demonstrating its robustness. Notably, most prior works (except NAYER) did not report such statistics, and due to their high computational cost, we were unable to reproduce their results.

Table 18: Averaging accuracy and standard deviation in three runs.

|  | CIFAR10 | | | CIFAR100 | | |
|---|---|---|---|---|---|---|
|  | R34/R18 | W402/W162 | W402/W161 | R34/R18 | W402/W162 | W402/W161 |
| SpaceshipNet | **95.39** | 93.25 | 90.38 | 77.41 | 69.95 | 58.06 |
| NAYER | $95.21 \pm 0.15$ | $94.11 \pm 0.18$ | $91.94 \pm 0.15$ | $77.56 \pm 0.12$ | $71.72 \pm 0.14$ | $62.23 \pm 0.21$ |
| K-Gen-S | $95.25 \pm 0.12$ | $94.12 \pm 0.13$ | $92.11 \pm 0.09$ | $77.58 \pm 0.10$ | $72.01 \pm 0.16$ | $62.78 \pm 0.17$ |
| K-Gen | $95.28 \pm 0.11$ | $94.19 \pm 0.11$ | $92.20 \pm 0.14$ | $77.70 \pm 0.09$ | $72.15 \pm 0.18$ | $62.84 \pm 0.20$ |

# F ADDITIONAL RESULTS ON X-RAY DATASET ($3000 \times 3000$ PIXELS)

To further assess the generalizability of K-Gen under large domain shift, we evaluate it on a chest X-ray classification task using a ResNet-18 teacher that attains 71.96% accuracy. In that we keep all hyperparameter as similar with the experiments in ImageNet. As reported in Table 19, K-Gen consistently outperforms both Fast100 and NAYER across 10% and 20% data memory ratios, substantially narrowing the gap to the full-data teacher despite using only a fraction of the original images. We also vary the distilled image resolution from $96 \times 96$ to $112 \times 112$ and $128 \times 128$, and observe that K-Gen remains stable and competitive across these settings, without re-tuning the loss weights. These results indicate that K-Gen is robust not only to significant domain shift from natural images to medical X-rays, but also to moderate changes in spatial resolution, supporting its applicability to real-world medical imaging scenarios.

Table 19: Results on X-ray dataset ($3000 \times 3000$ pixels) Karargyris et al. (2021).

| Method | X-ray Classification | |
|---|---|---|
|  | **10%** | **20%** |
| Teacher/Student | ResNet-18 (71.96) / ResNet-18 (71.96) | |
| Fast100 | 48.73 | 55.10 |
| NAYER | 52.84 | 58.92 |
| K-Gen | **66.15** | **69.84** |

# G FURTHER DISCUSSION

**Choosing Target Mask** $M_{target}$**.** In this section, we compared the performance of different target masks ($M_{\text{target}}$) across various sampling ratios (1%, 5%, 10%, and 20%). The target masks include Full($n$), where the matrix is filled with the value $n$, and G($i, j$), representing Gaussian matrices with a maximum value of $i$ and a standard deviation of $j$. As shown in Table 20, the "G(1,2)" matrix consistently outperforms other configurations, achieving the highest accuracy at all sampling ratios. While the "Full(1)" and "G(1,3)" matrices exhibit similar performance, they are generally outperformed by "G(1,2)" at most ratios. This indicates that gaussian the matrix is the most effective approach for this task.

Table 20: Performance Comparison Between Different Target Mask $M_{target}$. In that, Full(n) indidate matrix is fill by n and G(i,j) mean the Gaussian Matrix with max value of $i$ and $\sigma = j$

| Ratio | G(1,2) | G(1,3) | G(2,2) | G(2,3) | G(3,2) | G(3,3) | Full(1) | Full(2) | Full(3) |
|---|---|---|---|---|---|---|---|---|---|
| 1% | **34.52** | 34.4 | 33.11 | 34.32 | 34.26 | 33.3 | 33.3 | 34.49 | 33.94 |
| 5% | **80.32** | 79.99 | 78.68 | 80.11 | 79.39 | 79.52 | 78.6 | 79.67 | 79.96 |
| 10% | **86.67** | 86.53 | 86.24 | 86.31 | 86.44 | 85.7 | 86.56 | 86.12 | 85.88 |
| 20% | **88.25** | 88.11 | 87.63 | 88.25 | 87.38 | 87.45 | 88.07 | 87.84 | 87.85 |

**Choosing Class Representative Embedding** $f_y$**.** We evaluate the impact of using Label Text Embedding (LTE) and Class Center (CC) as the Class Representative Embedding $f_y$. The results in Table 21 show that K-Gen consistently outperforms NAYER across all settings. Furthermore, the performance of LTE and CC is comparable, with LTE exhibiting a slight advantage in some cases. This demonstrates the effectiveness of both configurations, providing flexibility in selecting between Class Center and Label Text Embedding representations.

Table 21: Performance comparison of K-Gen (using Class Center (CC) and Label Text Embedding (LTE)).

| Dataset | Imagenetee | | | Imagewoof | | |
|---|---|---|---|---|---|---|
| Teacher | Resnet34 (94.06) | | | Resnet34 (83.02) | | |
| Student | Resnet18 (93.53) | | | Resnet18 (82.59) | | |
| Ratio | 1% | 5% | 10% | 1% | 5% | 10% |
| NAYER | 9.35 | 32.17 | 42.57 | 6.72 | 15.62 | 25.27 |
| K-Gen (CC) | 34.43 | 80.22 | 86.43 | 20.35 | 36.21 | 59.35 |
| K-Gen (LTE) | 34.51 | 80.36 | 86.61 | 20.47 | 36.41 | 59.62 |

## H  VISUALIZATION

Figure 5 shows synthetic images generated by NAYER (a) at $224 \times 224$ and K-Gen (b) at $112 \times 112$, both after 100 generator training steps on ImageNet using ResNet-50 as the teacher. While challenging for human recognition and differing from real datasets, K-Gen's lower-resolution images capture key class-level features, showing superior quality over NAYER Tran et al. (2024b). In (c), the CAM for K-Gen's images reveals high CAM ratios across most pixels, highlighting the benefit of Key Region Loss.

In addition, Figure 6 provides further qualitative examples of K-Gen on randomly selected ImageNet classes at $112 \times 112$ resolution. Across a wide variety of object categories, the generated images remain low-resolution and abstract, yet consistently preserve class-defining structures (e.g., characteristic shapes, silhouettes, and textures), while backgrounds and non-discriminative regions vary more freely. Together with Figure 5, these visualizations support our claim that K-Gen concentrates generative capacity on key regions that are most relevant for the teacher, enabling compact synthetic datasets that still convey rich class-level information.

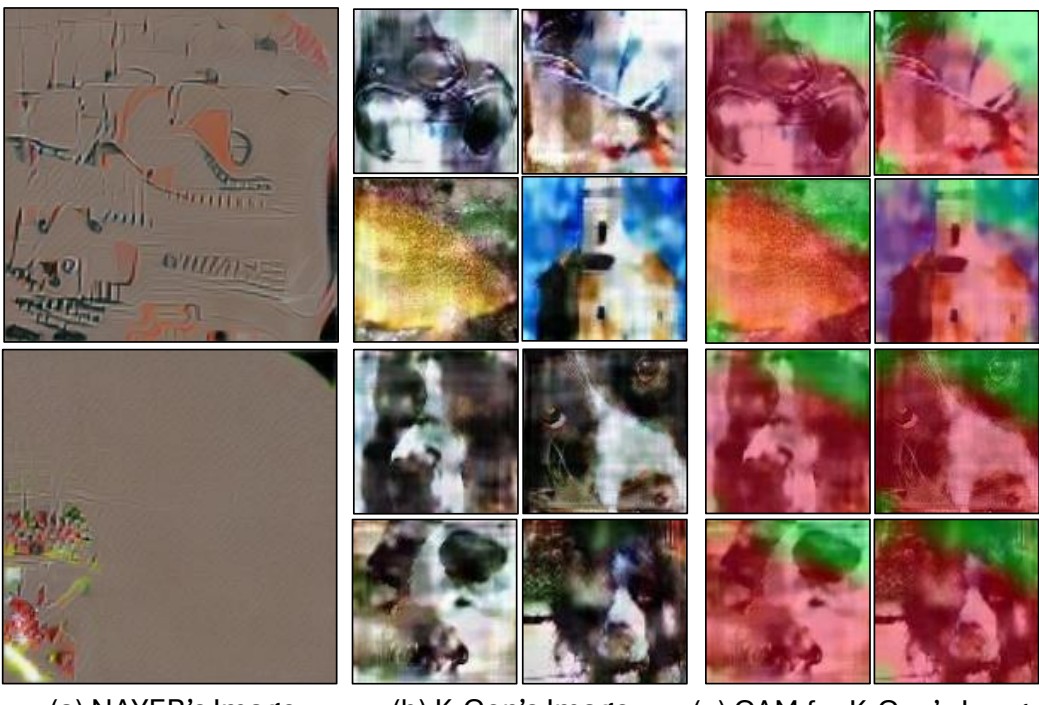

(a) NAYER's Image        (b) K-Gen's Image        (c) CAM for K-Gen's Image

Figure 5: (a-b) Synthetic data generated from the 'cassette player', 'tench', 'church', and 'English springer' classes of ImageNet1k, with NAYER (at $224 \times 224$ resolution) and our K-Gen (at $112 \times 112$ resolution). (c) Class activation map for our K-Gen's images. Please note that the values of the class activation map are shown before normalization.

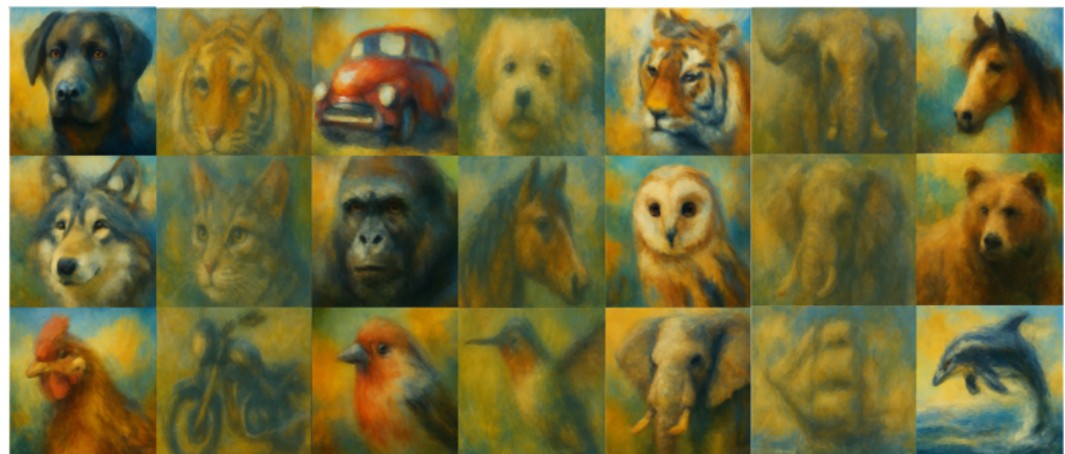

Figure 6: Further visualizations of our K-Gen on randomly selected ImageNet classes at a resolution of $112 \times 112$ pixels.

## I  FUTURE WORK

Our paper employs a customized version of the classic CAM, designed to facilitate backpropagation in obtaining the activation matrix. This approach opens the door to exploring other techniques, such as Grad-CAM Selvaraju et al. (2017) or attention-based scores Leem & Seo (2024), to further enhance the task. Additionally, optimizing multi-resolution techniques for faster processing times presents another promising direction for improvement.

## J  THE USE OF LARGE LANGUAGE MODELS

We used a large language model (ChatGPT) to help with editing this paper. It was only used for simple tasks such as fixing typos, rephrasing sentences for clarity, and improving word choice. All ideas, experiments, and analyses were done by the authors, and the use of LLMs does not affect the reproducibility of our work.

