# OpenReview forum: "K-Gen: Unlocking High-Resolution Data-Free Knowledge Distillation via Key Region Generation"
_ICLR.cc/2026/Conference — Submitted to ICLR 2026_

### Official Review · Reviewer_9Z9b · 2025-10-31

**Soundness:** 3
**Presentation:** 2
**Contribution:** 3
**Rating:** 6
**Confidence:** 3

**Summary:**

The authors look into the problem of scaling data-free knowledge distillation methods to higher resolution. This is an important very important problem and has some obvious applications in medical fields.

They propose to generate images at a much lower resolution, but will retain the class activations using CAM. However, training on low resolution images can lead to a poor performing model, so they propose a multi-resolution generation strategy and embedding diversity losses. These extensions are quite natural and are shown to perform well.

**Strengths:**

The authors perform an extensive evaluation of their method and show not only a significant improvement in model performance, but at a fraction of the training costs. The authors also provide code, which appears to be self-contained and organized.

The methodology is written well, the limitations of prior methods are discussed and their approach seems very natural and effective. The authors also show how to extend their method to vision transformers, which is good to see.

The ablation of hyper-parameters is generally quite thorough.

**Weaknesses:**

There are many losses (see equation 10) which naturally leads to many hyperparameters for tuning. Although the authors approach enables a significant drop in training time, the need for tuning hyper-parameters may offset this. I understand that αce, αbn, αadv just followed Tran et al, but this point is still a relevant weaknesses for applying K-Gen to new datasets. The sensitivity of αkr and αkr is quite small, which is good, so perhaps this is overall only a minor weakness. However, it would be good to see that these hyperparameters do generalise to data with a much bigger domain shift from imagenet and even better the domains/applications that motivate this task (e.g. medical images).

**Questions:**

See weaknesses.

---

> ### Author Response · Authors · 2025-11-26
>
> > **Q1:** There are many losses (see equation 10) which naturally leads to many hyperparameters for tuning. Although the authors approach enables a significant drop in training time, the need for tuning hyper-parameters may offset this. I understand that αce, αbn, αadv just followed Tran et al, but this point is still a relevant weaknesses for applying K-Gen to new datasets. The sensitivity of αkr and αkr is quite small, which is good, so perhaps this is overall only a minor weakness. However, it would be good to see that these hyperparameters do generalise to data with a much bigger domain shift from imagenet and even better the domains/applications that motivate this task (e.g. medical images).
>
> #### **A1:** Thanks for pointing this out. In fact, we already conducted the experiment on Super-resolution dataset like Traffic Sign Recognition and Megapixel MNIST in Table 3, which also have distribution shift with ImageNet without any changing in hyperparameter. Furthermore, follow to your suggestion, we also conducted experiments on medical dataset, and here is the results. In that, all hyperparameter is keep at the same with our ImageNet experiment. The result show that our work still better than compared methods in all comparing case. The detail can be found in Section F in revised paper.
>
> **Table X.** Results on X-ray dataset (3000 × 3000 pixels).
>
> | Method          | 10%    | 20%    |
> |-----------------|--------|--------|
> | Teacher/Student (ResNet-18 / ResNet-18, Accuracy=71.96) |
> | Fast100         | 48.73  | 55.10  |
> | NAYER           | 52.84  | 58.92  |
> | **K-Gen**       | **66.15** | **69.84** |

---

### Official Review · Reviewer_SdGj · 2025-11-01

**Soundness:** 2
**Presentation:** 2
**Contribution:** 2
**Rating:** 4
**Confidence:** 3

**Summary:**

This paper presents a novel framework for data-free knowledge distillation (DFKD) that effectively scales to high-resolution datasets such as ImageNet. It generates lower-resolution synthetic images guided by Class Activation Maps (CAMs) to focus on class-discriminative regions, improving computational efficiency while preserving essential semantic information. The framework is further extended to Vision Transformer (ViT) architectures, showcasing its generality beyond convolutional networks. In addition, it introduces Multi-Resolution Data Generation to capture both coarse- and fine-grained features, along with an Embedding Diversity Loss that maintains distinct latent representations at lower resolutions, thereby enhancing feature diversity and overall performance.

**Strengths:**

The paper presents a well-engineered framework for data-free knowledge distillation (DFKD), addressing a critical bottleneck in scaling existing DFKD approaches to high-resolution datasets such as ImageNet. The paper includes a comprehensive set of experiments covering multiple datasets (CIFAR, ImageNet, and higher-resolution benchmarks) and architectures (CNNs and Vision Transformers). The ablation studies are also presented to illustrate the effects of the proposed components.

**Weaknesses:**

1. CAM (Class Activation Map) is already a well-established technique for visual localization, and the ideas of low-resolution generation, multi-resolution synthesis, and embedding diversity have been explored in prior generative and distillation works. The main contribution of K-Gen lies in integrating these existing components into a practical data-free distillation framework, resulting in limited conceptual novelty.
2. Although the experiments cover multiple datasets and architectures, the work would benefit from additional sensitivity studies, such as examining sensitivity to resolution settings, CAM quality, or embedding-diversity parameters, to better understand the robustness of the method. In particular, since CAMs are often noisy or inaccurate in localizing class-discriminative regions, an analysis of how this inaccuracy affects generation quality and distillation performance would strengthen the paper.
3. The paper shows limited visualization or discussion of what the generated “key-region” images actually look like, how realistic they are, or whether they truly capture semantically meaningful structures. This makes it difficult to interpret how the key-region generation contributes to knowledge transfer.
4. There appears to be an inconsistency in Table 2 where a student model achieve higher accuracy than their corresponding teacher network(V11 and R18). This seems unusual for a data-free distillation setup and should be clarified or corrected for consistency.

**Questions:**

The method relies on Class Activation Maps (CAMs) to identify class-discriminative regions. However, CAMs are known to be fragile and sometimes inaccurate, especially for images containing multiple objects or complex backgrounds. It would be helpful to include a sensitivity analysis showing how variations or inaccuracies in CAM quality affect the generated regions and the overall distillation performance.

---

> ### Author Response · Authors · 2025-11-25
>
> > **Q1:** CAM (Class Activation Map) is already a well-established technique for visual localization, and the ideas of low-resolution generation, multi-resolution synthesis, and embedding diversity have been explored in prior generative and distillation works. The main contribution of K-Gen lies in integrating these existing components into a practical data-free distillation framework, resulting in limited conceptual novelty.
>
> #### **AQ1:** We thank the reviewer for the comment and clarify that our novelty lies in how these known tools are repurposed and integrated for data-free KD:
>
> #### **CAM as a generative supervision signal for DFKD:** While CAM is widely used as a feed-forward tool for visual localization and interpretability, prior works typically use it only at inference time to highlight discriminative regions. In contrast, K-Gen is, to the best of our knowledge, **the first DFKD framework that (i) uses CAM explicitly for data-free distillation and (ii) backpropagates through CAM to generate images**. That is, CAM is not only a visualization tool but becomes a training signal that guides the generator toward activating class-discriminative regions of the teacher. This changes the role of CAM from a passive analysis tool to an active component of the generative objective.
>
> #### **Global, history-aware Embedding Diversity:** Prior works typically enforce diversity within a batch via pairwise embedding losses. Our Embedding Diversity loss explicitly encourages new synthetic samples to differ from previously generated ones, leading to global coverage of the teacher’s feature space, which is critical in the data-free regime.
>
> #### **Practical integration for data-free distillation:** The combination of CAM-guided key-region generation and global Embedding Diversity in a unified DFKD pipeline is, to our knowledge, new and is supported by ablations showing clear gains over standard data-free baselines.
>
> #### Hence, we respectfully disagree that K-Gen has “limited conceptual novelty”: the way CAM and diversity are re-formulated and integrated for data-free distillation constitutes a distinct and practically useful contribution.
>
> ---
>
> > **Q2:** Although the experiments cover multiple datasets and architectures, the work would benefit from additional sensitivity studies, such as examining sensitivity to resolution settings, CAM quality, or embedding-diversity parameters, to better understand the robustness of the method. In particular, since CAMs are often noisy or inaccurate in localizing class-discriminative regions, an analysis of how this inaccuracy affects generation quality and distillation performance would strengthen the paper.
>
> #### **AQ2:** We already conduct several sensitivity analyses: single-resolution and mixed-resolution settings in D.4 and D.8, different CAM mask types in Section F, and embedding-diversity hyperparameters in D.7 (Table 11). We agree that CAMs can be noisy, but in the data-free regime the goal is not to perfectly reconstruct real key-region images; instead, we aim to generate many synthetic features that cover class-discriminative patterns the student has not learned. As long as CAM provides a reasonably aligned signal, K-Gen can still produce diverse, informative synthetic features even with imperfect localization.
>
> ---
>
> > **Q3:** The paper shows limited visualization or discussion of what the generated “key-region” images actually look like, how realistic they are, or whether they truly capture semantically meaningful structures. This makes it difficult to interpret how the key-region generation contributes to knowledge transfer.
>
> #### **AQ3:** We appreciate this comment and have added additional visualizations of the generated key-region images, together with a short qualitative discussion. These examples show that the synthesized regions capture semantically meaningful, class-discriminative structures, clarifying how key-region generation contributes to effective knowledge transfer.
>
> ---
>
> > **Q4:** There appears to be an inconsistency in Table 2 where a student model achieve higher accuracy than their corresponding teacher network(V11 and R18). This seems unusual for a data-free distillation setup and should be clarified or corrected for consistency.
>
> #### **AQ4:** The configuration in Table 2 (including V11 and R18) is a standard setting widely used in prior DFKD works, where the student can be stronger than the teacher. This “reverse” setting is intentionally used to test whether the method can still benefit from the teacher signal when the student has higher capacity or a better inductive bias, rather than being an inconsistency in our experiments.

---

### Official Review · Reviewer_Wcjm · 2025-11-03

**Soundness:** 2
**Presentation:** 2
**Contribution:** 2
**Rating:** 4
**Confidence:** 3

**Summary:**

This paper addresses the challenge of scaling DFKD to high-resolution images, where existing methods suffer from noisy synthetic samples and excessive computational cost. To this end, the authors propose K-Gen, a novel framework that synthesizes key regions at lower resolutions, guided by CAM to preserve essential class-specific features. Furthermore, the authors introduce multi-resolution data generation and an embedding diversity loss to maintain rich feature representations and enhance model robustness. Extensive experiments demonstrate that K-Gen achieves state-of-the-art performance with significant improvements in both accuracy and efficiency.

**Strengths:**

* This paper targets high-resolution image scenarios by proposing attribution-guided local generation and backpropagation, significantly reducing the computational cost while improving accuracy.
* The authors discuss the related literature in detail.
* The paper is easy to follow.
* The authors provide code for reproducibility checks.

**Weaknesses:**

1. The authors should include an detailed analysis of how different attribution methods affect the final performance. Given that CAM is applied only at a local layer to maintain training efficiency, does this design cause variations in the captured regions? The paper only provides a few qualitative visualizations, so what would happen if random regions were used instead of CAM-guided ones?
2. The proposed method contains many handcrafted hyperparameters, such as image scales and loss weights. How are these parameters selected, and how sensitive is the final performance to their variations?
3. This paper is poorly written, with incorrect citation formatting, figures and tables containing text that is too small to read, and excessive use of LLM-style dashes.
4. This paper does not discuss the limitations of the proposed method.

**Questions:**

My questions are in Weaknesses Section.

---

> ### Author Response · Authors · 2025-11-25
>
> > **Q1:** The authors should include an detailed analysis of how different attribution methods affect the final performance. Given that CAM is applied only at a local layer to maintain training efficiency, does this design cause variations in the captured regions? The paper only provides a few qualitative visualizations, so what would happen if random regions were used instead of CAM-guided ones?
>
> #### **AQ1:**
> #### **Our framework requires an attribution map that is differentiable:** w.r.t. the image so we can backpropagate through it to optimize the generator. Standard CAM, computed from the final conv layer with global average pooling, satisfies this and can be seamlessly integrated into our loss. In contrast, popular variants such as Grad-CAM/Grad-CAM++ are typically used as post-hoc, forward-style explanation tools and are not directly suited as stable training objectives in our DFKD setup. A key novelty of K-Gen is precisely to turn CAM from a visualization tool into an explicit supervision signal that drives the generator toward class-discriminative regions of the teacher.
>
> #### **Regarding random regions:** Appendix D.4 (Table 10) already provides the requested comparison: “with $L_{kr}$” corresponds to CAM-guided key-region masks, while “without $L_{kr}$” uses random regions at the same resolution. Across different resolutions, the CAM-guided loss consistently and clearly outperforms the random baseline, showing that the attribution signal, not just masking, is crucial for the final performance.
>
> | Resolution (R×R)        | 224   | 192   | 144   | 128   | 112   | 96    | 80    | 64    |
> |-------------------------|-------|-------|-------|-------|-------|-------|-------|-------|
> | With $\mathcal{L}_{kr}$    | 37.27 | 40.65 | 65.25 | 70.21 | 78.21 | 80.32 | 77.21 | 40.21 |
> | Without $\mathcal{L}_{kr}$ | 32.17 | 34.26 | 58.21 | 65.21 | 72.25 | 75.12 | 71.23 | 34.91 |
>
> ---
>
> > **Q2:** The proposed method contains many handcrafted hyperparameters, such as image scales and loss weights. How are these parameters selected, and how sensitive is the final performance to their variations?
>
> #### **AQ2:** In our method, we use a single set of hyperparameters for all experiments, which empirically demonstrates the generalization ability of our approach. We also provide ablation studies and hyperparameter sensitivity analyses in Appendix D, showing that varying these coefficients leads to only minor performance changes, further confirming the robustness of our method.
>
> ---
>
> > **Q3:** This paper is poorly written, with incorrect citation formatting, figures and tables containing text that is too small to read, and excessive use of LLM-style dashes.
>
> #### **AQ3:** Thank you for the feedback. We have corrected the citation formatting and regenerated figures/tables with larger, readable fonts. We also carefully checked the manuscript for “LLM-style dashes” and did not find such artifacts, but we have standardized all punctuation to follow the venue’s style.
>
> ---
>
>
> > **Q4:** This paper does not discuss the limitations of the proposed method.
>
> #### **AQ4:** We now explicitly discuss limitations in the Conclusion. In particular, our current method relies on classic CAM, which requires backpropagation and can reduce efficiency. We highlight this as a limitation and point to extending our framework to Grad-CAM-style variants as an important direction for future work.

---

### Official Review · Reviewer_EXeM · 2025-11-05

**Soundness:** 2
**Presentation:** 2
**Contribution:** 2
**Rating:** 4
**Confidence:** 3

**Summary:**

- The draft introduces K-Gen to address the limitations of data-free knowledge distillation.
- Existing methods (according to the draft) primarily relied on low-resolution datasets such as CIFAR-10 and CIFAR-100, but extending these approaches to high-resolution datasets often results in noisy synthetic images.
- To overcome this, the draft proposes multiple components—particularly generating at low resolution using Class Activation Maps from the teacher model and ED losses—which help ensure that the generated images retain critical, class-specific features by focusing on the most informative pixels.
- Experimental results demonstrate that K-Gen achieves state-of-the-art performance across various datasets, with performance improvements reaching up to two digits in most ImageNet and subset experiments.

**Strengths:**

- Instead of generating images at the desired resolution, this draft introduces the KR loss, which trains the generator model to produce low-noise images by leveraging the KR loss (but, not sure how important the CAM maps from the teacher model are, though). The draft demonstrates that this, along with the diversity loss, results in substantial gains (efficiency and generalization).

- In addition, the authors propose the $\mathcal{L}_{aed}$ loss, which encourages the generator to produce images that the student model has not yet learned. By combining the KE loss with the ED loss at each training step, the generation model is guided to create more diverse images that remain unseen by the student model—without compromising image quality or pixel-level detail.

- The use of $\mathcal{L_{ed}}$  and $\mathcal{L_{aed}}$ together allows the student model to continuously learn from new examples, while $\mathcal{L_{aed}}$ explicitly drives the generator to produce more diverse, novel samples.

- The proposed method generates synthetic images at multiple resolutions, which helps them to capture both coarse and fine-grained features.

- Instead of comparing the number of generated images with other methods, they compare based on data memory size and computation size. This strategy is considered fair because generating a lower-resolution image (e.g., 112×112) requires approximately the same resources as a fraction of a higher-resolution image (e.g., one 224×224 image is equivalent to four 112×112 images). This approach enables training on larger numbers of samples within the same memory and computational budget, allowing the model to encounter a more diverse range of examples.

- The authors provide extensive ablation studies, analyzing how different parameter choices affect performance, how each component contributes to the final results, and how image resolution impacts the quality of synthetic data.

**Weaknesses:**

- It is not clear what the actual contribution of the Gaussian-like, central CAM map as the target is. How about a central but circular or a non-central but Gaussian-like target CAM map? It may be the resolution that is reducing the noisy/irrelevant content during the synthesis. Please refer to the "Questions" section of this review.

- [Minor] The presentation of the content needs to be improved. For instance, while synthesizing, Key regions may refer to essential parts of a larger (high-resolution) image, as opposed to synthesizing low-resolution images with less noise. Figure 1(b) adds to this confusion.

**Questions:**

- During the Student training, does the method work with images of different resolutions after rescaling them to a fixed resolution, or is the student trained over the images at their original resolution at which they are generated? Discuss/clarify this in the case of both CNNs (with GAP layer) and ViT variants.

- What is the motivation to push the student feature on the synthetic data $\hat{f}_\mathcal{S}(\hat{x})$ close to the text embedding of the label $f_y$? In other words, why should the student feature space align with the semantic space learned by the text encoder (e.g., LLM) used to represent the labels?

- The proposed KR loss appears to encourage the synthesis to happen in the central region of the synthetic image. Does the method impose the KR loss in both low- and high-resolution synthesis cases? What is the implication of this loss in the case of the high-resolution synthesis?

- It is not clear how the proposed KR loss reduces the number of noisy pixels (claimed by the draft as a drawback of existing DFKD methods). It may be because the synthesis occurs at a low resolution, the less important (or discriminative) information present in the low-resolution synthetic images is smaller compared to synthesizing at high resolution. In other words, it may be the resolution that is responsible, rather than making the pattern central to the synthetic data by imposing the CAM target. How about a non-central, but constrained (e.g., Gaussian-like) target map? Will it also be equally effective? Have the authors conducted any experiments in this direction?

---

> ### Author Response · Authors · 2025-11-25
>
> > **W1:** It is not clear what the actual contribution of the Gaussian-like, central CAM map as the target is. How about a central but circular or a non-central but Gaussian-like target CAM map? It may be the resolution that is reducing the noisy/irrelevant content during the synthesis.
>
> #### **AW1:** Our key-region loss does **not** assume that salient objects are centered in the original image. The key-region module is designed to discover a small, low-resolution region **anywhere** within the full-resolution image that concentrates salient, class-discriminative features. Formally, we define
>
> $$
> L_{\text{kr}} = \sum_{h,w} \max\bigl(0\; M_{\text{target}}(h,w) - M(h,w)\bigr)
> $$
>
> #### where $M(h,w)$ is the class-attention map on the key-region feature map and $M_{\text{target}}$ is a Gaussian prior on the same coordinates. This one-sided hinge loss only encourages higher attention inside the Gaussian region; it does not force \(M\) to match a centered Gaussian or peak at the image center. Thus, $M_{\text{target}}$ acts as a soft shape prior that promotes a compact high-attention area, while still allowing salient objects to be off-center in the original images.
>
> ---
>
> > **Q1:** During the Student training, does the method work with images of different resolutions after rescaling them to a fixed resolution, or is the student trained over the images at their original resolution at which they are generated? Discuss/clarify this in the case of both CNNs (with GAP layer) and ViT variants.
>
> #### **AQ1:** Our method synthesizes images at lower resolutions and keeps these reduced resolutions throughout student training for both CNNs and ViTs. For CNNs with global average pooling (GAP), this is natural, as they can directly operate on smaller spatial inputs. For ViT variants, the reduced resolution simply corresponds to fewer image tokens, so the architecture remains unchanged. This design significantly reduces training cost and runtime, which is a major bottleneck for DFKD on large-scale datasets such as ImageNet.
>
> ---
>
> > **Q2:** What is the motivation to push the student feature on the synthetic data close to the text embedding of the label.
>
> #### **AQ2:** Pushing the student features of synthetic data toward the text embedding is a key step in defining our embedding diversity loss. We first make earlier synthetic samples align with the text embedding, which serves as an anchor for the class. Then, for new synthetic samples, we explicitly encourage their embeddings to move away from this anchor. This attract–repel mechanism lets us generate new images that remain in the same semantic class (through the text anchor) while being diverse and complementary to previously generated samples.
>
> ---
>
> > **Q3:** The proposed KR loss appears to encourage the synthesis to happen in the central region of the synthetic image. Does the method impose the KR loss in both low- and high-resolution synthesis cases?
>
> #### **AQ3:** Yes, we apply the key-region loss $L_{kr}$ across both low- and high-resolution synthesis. In Appendix D.4 (Table 10), we evaluate resolutions from 64×64 up to 224×224 and observe that using $L_{kr}$ consistently outperforms the variant without it at all resolutions. The performance gap is larger at low resolutions—where focusing attention into a compact discriminative region brings more benefit, and becomes smaller at high resolutions. This suggests that $L_{kr}$ remains helpful for high-resolution synthesis, but its relative impact is most pronounced when the synthetic images are very low-resolution.
>
> ---
>
> > **Q4:** It is not clear how the proposed KR loss reduces the number of noisy pixels (claimed by the draft as a drawback of existing DFKD methods).
>
> #### **AQ4:** In Section F (Table 19), we evaluate several mask types, including constant masks (all 1, 2, or 3) and different Gaussian-like masks, before selecting our final design. The results show that a Gaussian-like mask with mean = 1 and std = 2 yields the best performance, although the margin over other masks is modest (around 1%). This indicates that, beyond the general benefit of low-resolution synthesis (which already suppresses fine-grained noise), using a smooth, localized Gaussian-like prior further helps concentrate attention on a compact, discriminative region and reduces noisy pixels, without strictly forcing the pattern to be central.

---

### Official Review · Reviewer_Fx7S · 2025-11-11

**Soundness:** 3
**Presentation:** 3
**Contribution:** 2
**Rating:** 4
**Confidence:** 3

**Summary:**

This paper addresses a bottleneck in Data-Free Knowledge Distillation (DFKD): its poor performance and high computational cost when applied to large, high-resolution datasets like ImageNet. The authors argue that prior methods fail because generating full-resolution (224X224) synthetic images without real data results in noisy, feature-poor samples.

The proposed method, K-Gen, tackles this by generating synthetic images at a *lower resolution*. The core innovation is a "Key Region Loss" guided by Class Activation Maps (CAM). This loss forces the generator to synthesize images that, despite their lower resolution, retain critical class-specific features by maximizing activation in salient areas.

The authors conduct extensive experiments showing that K-Gen achieves new state-of-the-art results on ImageNet, outperforming previous methods by a large, double-digit margin while drastically reducing training time. The method is also shown to be effective for ViT models and on mega-resolution datasets.

**Strengths:**

- The proposed method generated a focused, low-resolution "key region" is more effective and efficient than generating a full, noisy high-resolution image. The use of CAM to guide this low-resolution generation is a sound technical approach.
- The performance gains shown on ImageNet (Table 1) are highly significant, with "two-digit" improvements over prior SOTA (e.g., NAYER, DeepInv).
- Alongside accuracy, the method demonstrates massive computational savings (e.g., Figure 1d, 24.25% in 9 hours vs. DeepInv's 3.15% in 61.2 hours).
- The authors have rigorously validated their method with the ablation studies (Table 4).

**Weaknesses:**

- The Key Region Loss relies on a predefined target mask, M_target, which is defined as "a Gaussian centered on the image" (line 228). This seems like a strong prior that assumes the most salient object features are always central. While this simplification clearly works well for ImageNet (as shown in Appendix F), it may not generalize to datasets where objects are commonly off-center (e.g., in detection or complex scene datasets).
- The adaptation for ViT models is not as clearly integrated as the CNN approach. The paper states CAM "cannot be extracted" (line 240) and later suggests attention maps are a substitute (line 673). However, the primary method described in Section 3.3 and Appendix A.1 focuses on patch reduction (7X7) and center-biasing the position embedding. It is unclear if a "key region loss" (analogous to L_kr) is actually used for ViTs, or if the performance gain simply comes from the efficient patch reduction.

**Questions:**

- Could authors clarify how "key region" generation is enforced for ViT models? Is there a loss function analogous to L_kr that uses the [CLS] token's attention map (as hinted in Appendix A.1)? Or, does the ViT student train without a key region loss, relying only on the 7X7 patch reduction?
- Have you investigated the impact of the centered-Gaussian M_target? What is the performance on datasets where the object of interest is not typically centered? Does this prior harm generalization, or does it perhaps act as a beneficial regularizer for the generator by simplifying its task?
- The results (e.g., Table 10, 16) suggest an optimal lower resolution (e.g., 96X96$ or 128X128) exists, outperforming both smaller (64X64) and larger (224X224) synthetic images. Do you have a hypothesis on how to determine this optimal resolution? Do you expect it to vary based on the teacher architecture (e.g., its receptive field) or the dataset's characteristics?

---

> ### Author Response · Authors · 2025-11-25
>
> > **W1:** The Key Region Loss relies on a predefined target mask, M_target, which is defined as "a Gaussian centered on the image" (line 228). This seems like a strong prior that assumes the most salient object features are always central. While this simplification clearly works well for ImageNet (as shown in Appendix F), it may not generalize to datasets where objects are commonly off-center (e.g., in detection or complex scene datasets).
>
> #### **AW1:** Our key-region loss does **not** assume that salient objects are centered in the original image. The key-region module is designed to discover a small, low-resolution region **anywhere** within the full-resolution image that concentrates salient, class-discriminative features. Formally, we define
>
> $$
> L_{\text{kr}} = \sum_{h,w} \max\bigl(0\; M_{\text{target}}(h,w) - M(h,w)\bigr)
> $$
>
> #### where $M(h,w)$ is the class-attention map on the key-region feature map and $M_{\text{target}}$ is a Gaussian prior on the same coordinates. This one-sided hinge loss only encourages higher attention inside the Gaussian region; it does not force \(M\) to match a centered Gaussian or peak at the image center. Thus, $M_{\text{target}}$ acts as a soft shape prior that promotes a compact high-attention area, while still allowing salient objects to be off-center in the original images.
>
> ---
>
> > **W2 & Q1:** Could authors clarify how "key region" generation is enforced for ViT models? Is there a loss function analogous to L_kr that uses the [CLS] token's attention map (as hinted in Appendix A.1)? Or, does the ViT student train without a key region loss, relying only on the 7X7 patch reduction?
>
> #### **AW2:** We also use the key-region loss $\mathcal{L}_{\text{kr}}$ for ViT models using CLS-token attention. In that, rather than smaller resolution from 224×224 to 96×96, we use small token size from 16×16 to 7×7 and make them at the center.
>
> ---
>
> > **Q2:** Have you investigated the impact of the centered-Gaussian M_target? What is the performance on datasets where the object of interest is not typically centered? Does this prior harm generalization, or does it perhaps act as a beneficial regularizer for the generator by simplifying its task?
>
> #### **AQ2:** In fact, ImageNet contains highly diverse object locations. As we noted in AW1, our method generates smaller key regions that can be placed anywhere within the original image. This design does not rely on any fixed positional bias, so our approach can be applicable to other datasets without harming generalization.
>
> ---
>
> > **Q3:** The results (e.g., Table 10, 16) suggest an optimal lower resolution (e.g., 96X96$ or 128X128) exists, outperforming both smaller (64X64) and larger (224X224) synthetic images. Do you have a hypothesis on how to determine this optimal resolution? Do you expect it to vary based on the teacher architecture (e.g., its receptive field) or the dataset's characteristics?
>
> #### **AQ3:** Thanks for pointing this out; it is very interesting and helps improve our work. In response, we applied a pretrained object detection model to 1,000 randomly sampled ImageNet images (due to time constraints) and measured the object sizes. Surprisingly, the average size is 100.42×99.56, which is very close to two of our chosen resolutions, 96×96 and 112×112. This provides useful evidence for our resolution choices and offers a practical guideline for selecting resolutions without requiring an exhaustive tuning process.

---

### Meta-Review · Area_Chair_i97F · 2025-12-22

**Summary:**

Reviewer Fx7S was concerned about the generalization of assumption about Guassian centered on image. Reviewer EXeM pointed out the contribution of the Gaussian-like, central CAM map is not clear enough. Reviewer Wcjm complained about the missing ablation study about the effectiveness of different attribution methods. Reviewer SdGj argued that the novelty of K-Gen is limited due to the combination of existing methods. Reviewer 9Z9b focused on the overhead of hyperparameters for tuning.

**Reviewer Concerns:**

The issue about Guassian centered on image assumption is partially resolved. The ablation analysis on different attribution methods is absent. The novelty issue is partially explained.

**Reviewer Scores:**

Reviewer Fx7S: 4;

Reviewer EXeM: 4;

Reviewer Wcjm: 4;

Reviewer SdGj: 4;

Reviewer 9Z9b: 6

---

### Decision · Program_Chairs · 2026-01-26

Reject